# TurboGS: Accelerating 3D Gaussian Splatting via Error-Guided Sparse Pixel Sampling and Optimization

**Zheng Dong†** [1]  **Daifei Qiu** [2]  **Pinxuan Dai** [3]  **Ke Xu** [4]  **Jiamin Xu** [5]  **Lili He** [1]  **Rynson W.H. Lau** [4]  **Weiwei Xu†** [3]

## Abstract

Consumer-level applications require fast optimization of 3D Gaussian Splatting (3DGS) with high-fidelity novel view rendering. However, existing 3DGS acceleration approaches still incur substantial computation on redundant pixels while sacrificing fine details. In this paper, we present TurboGS, an error-guided training framework that accelerates 3DGS by concentrating optimization on perceptually informative pixels. TurboGS is built upon four core components: **(1)** a tile-wise sparse pixel sampling, which, driven by multi-view reconstruction errors during training, prioritizes challenging regions and skips well-reconstructed ones to avoid redundant gradient computation; **(2)** a tile-wise structure-aware loss with sparse Normalized Cross-Correlation, which provides sparse yet effective supervision to preserve fine details and stabilize training; **(3)** an error-driven Gaussian density control strategy, which dynamically allocates model capacity and removes redundant primitives; and **(4)** a tailored hybrid optimizer that couples Hessian-informed updates with Adam moment damping to stabilize and improve convergence under sparse supervision. Experiments on standard benchmarks demonstrate that TurboGS can deliver on par or superior rendering quality within 100 seconds (up to ∼10× training speedup over vanilla 3DGS).

## 1. Introduction

Novel view synthesis (NVS) is a long-standing and challenging research topic with downstream applications such as VR/AR. In recent years, Neural Radiance Fields (NeRF) (Mildenhall et al., 2020) and their variants demon-

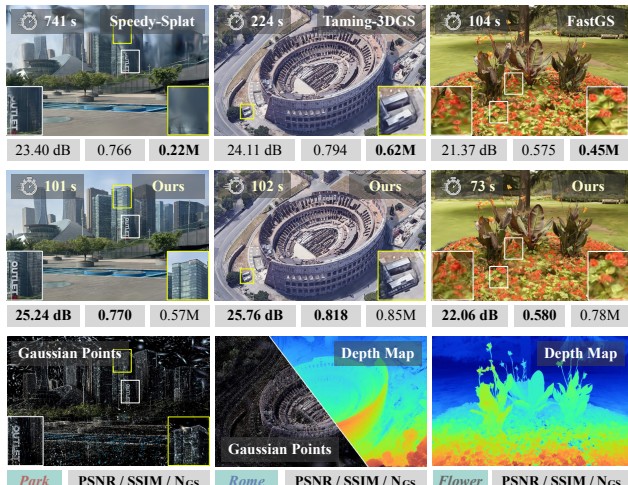

*Figure 1.* We propose **TurboGS**, a novel framework that *accelerates 3DGS optimization* while delivering high-quality novel view rendering results with notable details across scene scales.

strate impressive rendering fidelity, yet require hours to days of per-scene optimization. Despite substantial progress on acceleration (Müller et al., 2022; Hedman et al., 2021), NeRF-style approaches still struggle to simultaneously balance optimization time, memory footprint, and rendering quality of high-frequency details. On the other hand, by representing scenes with explicit Gaussian primitives, 3D Gaussian Splatting (3DGS) (Kerbl et al., 2023) offers fast optimization and rendering that achieves NeRF-comparable quality. Nonetheless, to satisfy consumer-facing or time-sensitive real-world applications, accelerating 3DGS (*e.g.,* with tight computation budgets or under large-scale/fast-turnaround settings (Liu et al., 2024; Zhu et al., 2026)) remains highly desirable.

Existing 3DGS acceleration methods largely fall into two categories. A group of works accelerates the rendering and backpropagation procedures via improved rasterization (Hanson et al., 2025a; Feng et al., 2025) or near-second-order optimizers (Höllein et al., 2025; Lan et al., 2025). The other group of methods focuses on controlling the Gaussian primitive complexity (*i.e.,* adaptive density control) by scheduling the primitive growth or pruning redundant Gaussians (Mallick et al., 2024; Chen et al., 2025; Fang & Wang, 2024; Hanson et al., 2025b; Papantonakis et al., 2024b; Wang et al., 2024; Baranowski et al., 2026). De-

[1]Zhejiang Sci-Tech University [2]Malanshan Audio&Video Laboratory [3]CAD&CG Laboratory, Zhejiang University [4]City University of Hong Kong [5]Hangzhou Dianzi University. Correspondence to: Zheng Dong†, Weiwei Xu† <zhengdong@zstu.edu.cn, xww@cad.zju.edu.cn>.

*Proceedings of the 43$^{rd}$ International Conference on Machine Learning*, Seoul, South Korea. PMLR 306, 2026. Copyright 2026 by the author(s).

spite their success, as shown in Fig. 1(first row), existing approaches typically suffer from (1) substantial redundant computation in well-reconstructed regions due to their uniform optimization of pixels: rasterizing and backpropagating dense pixels/tiles per iteration during training; and (2) loss of fine details due to their aggressive primitive pruning.

To address these problems, we draw inspiration from the sparse ray sampling for optimizing NeRF (Mildenhall et al., 2020), where focusing on a set of informative rays improves efficiency. Analogously, we observe that during 3DGS optimization, residual errors are typically distributed across a small subset of difficult pixels (*e.g.*, edges, thin structures, and other high-frequency regions), while a large portion of pixels quickly becomes low-error and contributes marginal gradients. Repeated rasterization in such well-reconstructed regions may hinder convergence under a fixed time budget. To this end, we ask: *can we allocate optimization capacity adaptively according to the reconstruction difficulty, instead of uniformly over the whole image plane?*

In this paper, we propose **TurboGS**, a novel error-guided training framework that accelerates 3DGS by focusing optimization on perceptually informative pixels. We first propose tile-wise sparse pixel sampling (guided by online-maintained multi-view error maps) to prioritize per-iteration rasterization and gradient backpropagation on challenging regions. Based on the per-tile sparsely sampled pixels, we introduce a local structure-aware loss that stabilizes the optimization while preserving fine details, by measuring the sparse Normalized Cross-Correlation (NCC). We also propose an error-driven density control mechanism for adaptive Gaussian densification and pruning, which improves both efficiency and rendering quality by accumulating and leveraging per-Gaussian statistics corresponding to the sparse pixel errors. Last, we optimize TurboGS with a new, hybrid optimizer that blends Hessian-informed updates with Adam moment damping, to achieve stable and fast convergence.

Extensive experiments on standard benchmarks show that TurboGS delivers high-quality rendering results with fine details (see Fig. 1(second row)) and converges fast (see Fig. 2). In sum, we present the following main contributions:

- TurboGS, a novel error-guided training framework for 3DGS that accelerates optimization by dynamically focusing computation on challenging pixels.
- A tile-wise structure-aware loss based on sparse Normalized Cross-Correlation (NCC) to provide effective supervision from sparse samples, preserving fine details and local texture consistency.
- An error-driven density control strategy that enables targeted densification of challenging regions and pruning of redundant Gaussians.
- A tailored hybrid optimizer that integrates Adam moments with Hessian-informed updates, to enable faster and more

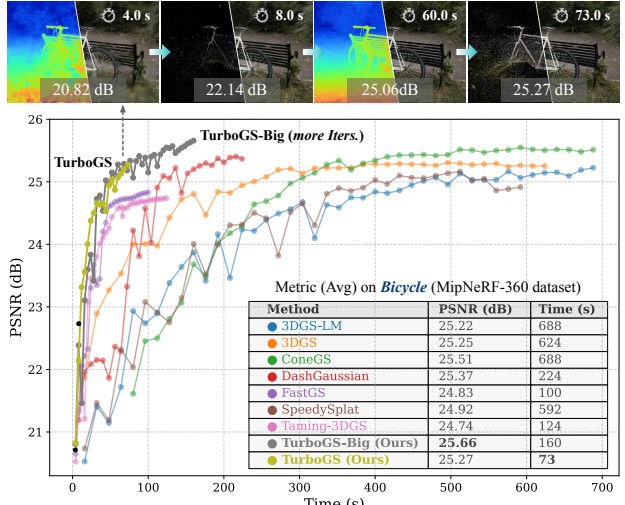

*Figure 2.* We visualize our intermediate test view rendering results (upper part) and training time-PSNR trajectories of existing methods and ours on the *Bicycle* scene (Yu et al., 2024) (lower part). Our method converges faster to better results.

stable convergence under challenging conditions of sparse, non-uniform gradients.

## 2. Related Work

**Novel View Synthesis** (NVS) attracts significant research interest in computer vision and graphics. Neural Radiance Fields (NeRF) (Mildenhall et al., 2020) significantly facilitates NVS by implicitly parameterizing scene information via an MLP network, and achieving high-quality image synthesis through volumetric rendering. While NeRFs suffer from training overheads, follow-up works combine NeRF with explicit or hybrid representations, such as voxels (Fridovich-Keil et al., 2022; Sun et al., 2022), hash grids (Müller et al., 2022), point-based formulations (Xu et al., 2022), to improve the training or rendering efficiency.

Alternatively, 3D Gaussian Splatting (3DGS) (Kerbl et al., 2023) represents scenes with explicit Gaussian primitives and renders images through GPU-friendly rasterization and $\alpha$-compositing, enabling much faster optimization and real-time rendering. Building upon 3DGS, prior works improve rendering quality (Hamdi et al., 2024; Lu et al., 2024; Yu et al., 2024), enhance surface reconstruction (Guédon & Lepetit, 2024; Huang et al., 2024), scale to large scenes or high-resolution rendering (Kerbl et al., 2024; Feng et al., 2025), and reduce reliance on SfM (Snavely et al., 2006) initialization (Ji & Yao, 2025; Pan et al., 2025; Fan et al., 2024b). Our TurboGS is inspired by NeRF-style sparse ray sampling, where we introduce a tailored principle to 3DGS by focusing supervision on informative pixels.

**Efficient 3DGS.** Recent methods aim to pursue higher efficiency in both training (*e.g.*, density control in Taming-3DGS (Mallick et al., 2024)) and rendering (*e.g.*, LoD

structures in Octree-GS (Ren et al., 2025)), while reducing the storage overhead induced by large numbers of primitives (Papantonakis et al., 2024a). In contrast, our TurboGS allocates the optimization budget over pixels based on reconstruction difficulty and training history.

*Primitive Management (Densification and Pruning).* A key bottleneck of 3DGS (Kerbl et al., 2023) is the rapid growth of Gaussian primitive count, yielding redundancy and slow optimization. Recent works control density by refining this schedule (Mallick et al., 2024; Hanson et al., 2025a; Chen et al., 2025; Kheradmand et al., 2024; Baranowski et al., 2026; Ren et al., 2026), developing stronger pruning criteria (Fan et al., 2024a; Rota Bulò et al., 2024; Fang & Wang, 2024), and combining compression with pruning for efficiency (Girish et al., 2024). For instance, Dash-Gaussian (Chen et al., 2025) schedules rendering resolution with primitive growth, FastGS (Ren et al., 2026) guides densification and pruning with multi-view reconstruction consistency, HGS (Xu et al., 2026) mines hard Gaussians from multi-view positional gradients, 3DGS-MCMC (Kheradmand et al., 2024) formulates MCMC with SGLD-style updates (Brosse et al., 2018; Shakiba Kheradmand, 2024), while Mini-GS (Fang & Wang, 2024) combines depth and blurriness cues with splitting and importance-aware pruning. In TurboGS, we back-project sparse multi-view pixel errors onto Gaussians during rasterization, where the aggregated error statistics guide targeted densification and pruning.

*Optimizer and Rasterization Acceleration.* Another group of recent works improves the efficiency of optimization and rasterization. 3DGS-LM (Höllein et al., 2025) introduces a two-stage Adam-to-LM optimization, while stochastic Newton-style prioritized per-kernel updates are used in 3DGS$^2$ (Lan et al., 2025) for fast convergence. To accelerate training, per-splat backward is used in Taming-3DGS (Mallick et al., 2024), while StopThePop (Radl et al., 2024), FlashGS (Feng et al., 2025), and Speedy-Splat (Hanson et al., 2025a) focus on Gaussian–tile pairs with accurate tile intersection. In TurboGS, we design an error-guided sparse pixel supervision with a lightweight hybrid solver, which enables more efficient optimization and better time–quality tradeoffs.

# 3. Our Proposed Method : TurboGS

## 3.1. Preliminary

**3D Gaussian Splatting (3DGS).** Given calibrated multi-view images $\{\mathbf{I}_v\}_{v=1}^N$ with corresponding cameras $\{\mathcal{C}_v\}$, and the point cloud derived from the structure-from-motion (SfM) (Snavely et al., 2006) algorithm, 3DGS represents a scene as a set of $M$ anisotropic Gaussian primitives $\mathcal{G} = \{g_i\}_{i=1}^M$. Each primitive $g_i$ is parameterized by geometry and appearance: a 3D center position $\boldsymbol{\mu}_i \in \mathbb{R}^3$, a 3D covariance $\boldsymbol{\Sigma}_i \in \mathbb{R}^{3 \times 3}$ (typically via scale $\mathbf{s}_i$ and rotation $\mathbf{R}_i$), an opacity $\alpha_i \in (0, 1)$, and color coefficients $\mathbf{c}_i$.

For a pixel $\mathbf{p}$ in view $v$, 3DGS projects Gaussians to the image plane and rasterizes them as 2D splats. Let $\mathcal{K}(\mathbf{p})$ be the ordered set of Gaussians intersecting pixel $\mathbf{p}$ (sorted by depth). The rendered color follows alpha compositing:

$$\hat{\mathbf{I}}_v(\mathbf{p}) = \sum_{k \in \mathcal{K}(\mathbf{p})} T_k(\mathbf{p}) \, \alpha_k(\mathbf{p}) \, \mathbf{c}_k(\mathbf{p}), \ \ T_k(\mathbf{p}) = \prod_{j<k} \big(1 - \alpha_j(\mathbf{p})\big),$$
(1)

where $\alpha_k(\mathbf{p})$ depends on the projected 2D Gaussian footprint and its opacity. This pipeline is differentiable, enabling gradient-based optimization of $\mathcal{G}$.

**Sparse-Pixel Training Setting.** Different from vanilla 3DGS (Kerbl et al., 2023) that rasterizes a dense image (or full tiles) for every selected view at each iteration, our **TurboGS** optimizes Gaussian primitives $\mathcal{G}$ using a sparse set of pixels to reduce redundant computation.

At iteration $t$, we first sample a small set of views $\mathcal{V}_t$ (Sec. 3.4). For each view $v \in \mathcal{V}_t$, we then sample a tile-wise sparse pixel set $\mathcal{P}_{t,v}$ (Sec. 3.5), where the sampled pixels are ordered by their tile indices to match the underlying rasterization schedule. TurboGS executes the standard 3DGS training loop only on these sampled pixels, including sparse forward rasterization, loss evaluation, and backpropagation.

To make sparse-pixel training efficient in practice, we introduce modifications to the original rasterization kernels via *tile-wise pixel mapping* and the *per-Gaussian backward* mechanism (Mallick et al., 2024), which restricts computation to the sampled pixel set, *i.e.,* $\{(v, \mathbf{p}) \mid v \in \mathcal{V}_t, \mathbf{p} \in \mathcal{P}_{t,v}\}$, while preserving tile-parallel forward execution and efficient splat-parallel computation in the backward pass.

## 3.2. Overview of TurboGS

Motivated by the observation that, as training progresses, only a small subset of pixels remains difficult to reconstruct while most become well explained and yield marginal gradients (see the decreasing pixel errors in Fig. 5), TurboGS maintains per-pixel statistics to allocate computation to informative pixels throughout training. As shown in Fig. 3 and Alg. 1, each iteration performs **(1)** *view selection* and **(2)** *tile-wise sparse pixel sampling* guided by pixel error and age, followed by **(3)** *sparse pixel rasterization*, **(4)** *tile-level sparse supervisions*, **(5)** *online EMA updates of error and age maps*, **(6)** *error-driven Gaussian density control*, and **(7)** *a lightweight hybrid solver* for stable sparse training.

## 3.3. Persistent Pixel-wise Informative Map

To support error-guided pixel sampling throughout training, TurboGS maintains *pixel-wise maps* for each training view. For each view $v$, we store (i) an **error map** $\mathbf{E}_v \in \mathbb{R}^{H \times W}$ that records an online estimate of per-pixel reconstruction difficulty, and (ii) an **age map** $\mathbf{A}_v \in \mathbb{N}^{H \times W}$ that measures how long a pixel has not been sampled. The error map

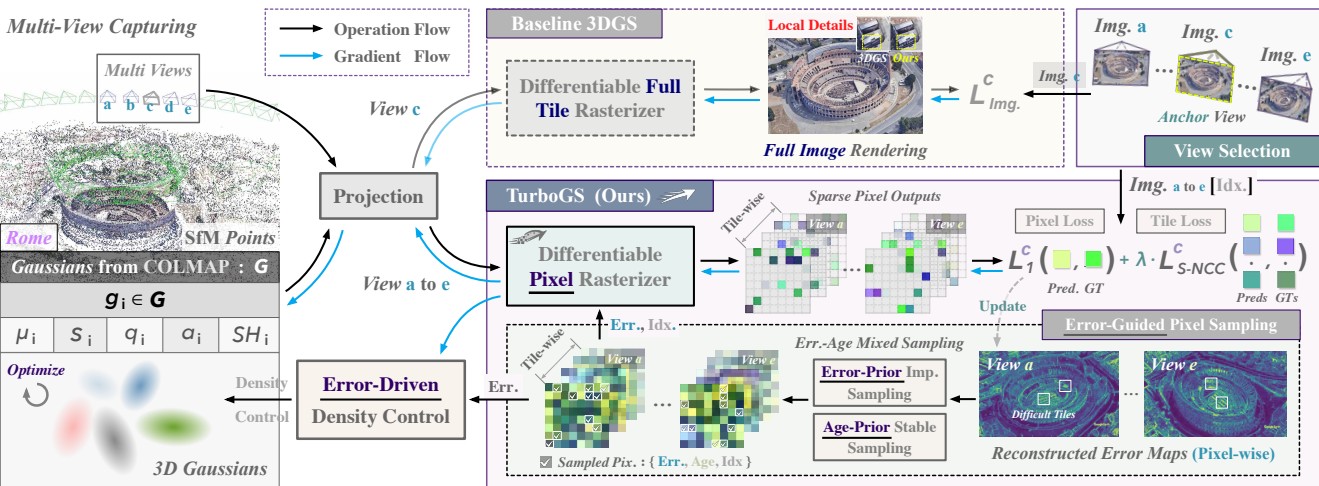

*Figure 3.* **TurboGS framework overview.** Unlike existing 3DGS (Kerbl et al., 2023; Mallick et al., 2024; Ren et al., 2026) methods that rasterize the entire tile in each iteration, TurboGS allocates more attention to pixels that are difficult to reconstruct and reduces gradient backpropagation in well-interpreted regions, thereby lowering the rasterization cost per iteration. To achieve this, TurboGS records multi-view pixel reconstruction errors and performs tile-wise sparse pixel sampling based on pixel error and age. Built upon this sparse sampling scheme, we further incorporate a local structure-aware loss and error-driven density control to preserve fine details.

promotes exploitation of difficult regions, while the age map encourages periodic revisiting of neglected pixels to refresh outdated estimates and to avoid over-focusing on a small set of persistently high-error locations.

**Sparse Pixel Loss as Error Signal.** For sampled pixels, we compute a per-pixel reconstruction error, as:

$$e_{t,v}(\mathbf{p}) = \left\| \hat{\mathbf{I}}_{t,v}(\mathbf{p}) - \mathbf{I}_v(\mathbf{p}) \right\|_1, \qquad \mathbf{p} \in \mathcal{P}_{t,v}, \quad (2)$$

and we initialize error map $\mathbf{E}_v$ by densely evaluating $e_{0,v}(\mathbf{p})$ for each pixel rendered by initial Gaussian $\mathcal{G}_0$.

**EMA Update of Pixel Errors.** We update $\mathbf{E}_v$ only at sampled locations using an exponential moving average:

$$\mathbf{E}_v(\mathbf{p}) \leftarrow (1 - \beta)\,\mathbf{E}_v(\mathbf{p}) + \beta\,e_{t,v}(\mathbf{p}), \qquad \mathbf{p} \in \mathcal{P}_{t,v}, \ (3)$$

and keep other pixels unchanged. This avoids re-estimating errors over all pixels at every iteration. Fig. 5 illustrates the error map $\mathbf{E}_v$ update process during training.

**Pixel Age Update.** We increment the age of all pixels globally and reset the age of sampled pixels:

$$\mathbf{A}_v(\mathbf{p}) \leftarrow \begin{cases} 0, & \mathbf{p} \in \mathcal{P}_{t,v}, \\ \mathbf{A}_v(\mathbf{p}) + 1, & \text{otherwise.} \end{cases} \quad (4)$$

In practice, we store $\mathbf{E}_v$ and $\mathbf{A}_v$ in contiguous GPU memories, and use `half` precision for $\mathbf{E}_v$ and `uint16` for $\mathbf{A}_v$.

### 3.4. Geometry-Aware View Sampling

At iteration $t$, we sample $K$ views to form a view set $\mathcal{V}_t$ As shown in Fig. 3 and Fig. 4, we first choose an *anchor view* according to the average error over its full pixels,

$$\bar{E}_v = \frac{1}{|\mathbf{\Omega}|} \sum_{\mathbf{p} \in \mathbf{\Omega}} \mathbf{E}_v(\mathbf{p}), \qquad v^\star = \arg\max_v \bar{E}_v, \quad (5)$$

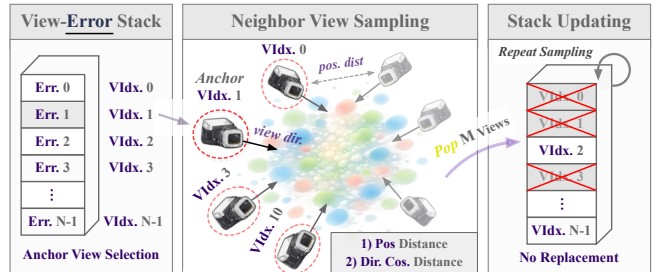

*Figure 4.* **View selection in TurboGS.** An anchor view is chosen based on view errors, followed by geometry-aware sampling of $K-1$ nearby views from a candidate stack. Selected views are then removed from the stack to avoid repeated sampling.

where $\mathbf{\Omega}$ denotes the image domain. We then sample the remaining $K-1$ views without repetition according to a geometry-aware distribution that favors nearby cameras with similar viewing directions to the selected anchor view:

$$\pi(u \mid v^\star) \propto \exp\!\Big(-\lambda_d \|\mathbf{c}_u - \mathbf{c}_{v^\star}\|_2 - \lambda_\theta \big(1 - \langle \mathbf{d}_u, \mathbf{d}_{v^\star} \rangle\big)\Big),$$
$$(6)$$

where $\mathbf{c}_u$ and $\mathbf{d}_u$ denote the camera center and unit viewing direction of the view $u$. In practice, we maintain a shuffled view `stack` and draw the $K$ views *without replacement* by popping sampled indices from the stack, which will be refilled and reshuffled once fewer than $K$ views remain.

### 3.5. Tile-Wise Error-Guided Sparse Pixel Sampling

We partition each image into several tiles of size $S \times S$ (*e.g.,* $16 \times 16$). For each selected view $v$ at iteration $t$, TurboGS samples a fixed number, *i.e.,* $N_{\text{pix}} = \max\big(1, \lfloor r_t \cdot S^2 \rfloor\big)$ of pixels *within each tile* to form $\mathcal{P}_{t,v}$, where $r_t$ denotes the pixel sampling rate. We further split $N_{\text{pix}}$ into a hard set

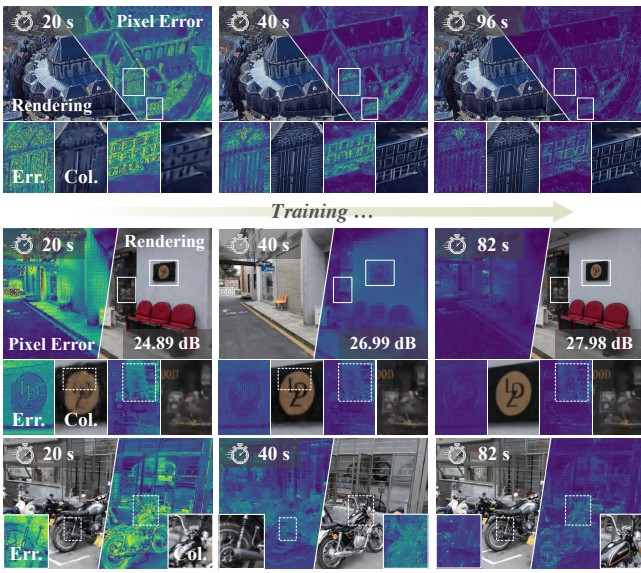

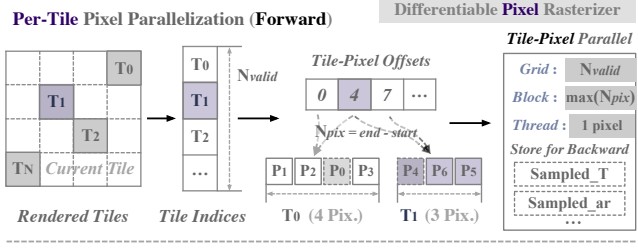

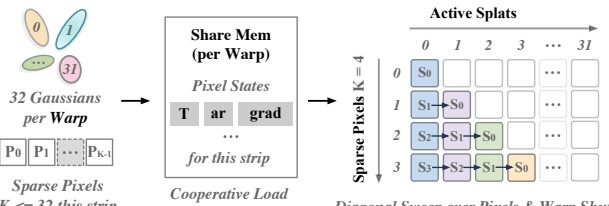

*Figure 6.* **Pixel rasterization in TurboGS.** Forward rasterization follows tile-parallel execution by organizing sampled pixels in tile order, while the backward pass adopts per-Gaussian splat parallelization to propagate gradients only over sparse sampled pixels.

*Figure 5.* **Visualization of error maps during training.** Pixelwise error maps and corresponding renderings over training (upper: *Amsterdam*; lower: *Street*). As optimization proceeds, pixel errors gradually weaken as reconstruction quality improves, and the error maps flatten toward predominantly low-frequency residuals.

and a stable set, with $N_{\text{hard}} = \lfloor r_{\text{hard}} \cdot N_{\text{pix}} \rfloor$, and $N_{\text{stable}} = \lfloor r_{\text{stable}} \cdot N_{\text{pix}} \rfloor$, satisfying $N_{\text{hard}} + N_{\text{stable}} \le N_{\text{pix}}$.

**Error-Prior Importance Sampling.** For a tile $\tau$, we define an error-based sampling distribution over its pixels:

$$\pi_E(\mathbf{p} \mid \tau, v) = \frac{\mathbf{E}_v(\mathbf{p})}{\sum_{\mathbf{q} \in \tau} \mathbf{E}_v(\mathbf{q}) + \epsilon}, \qquad \mathbf{p} \in \tau, \quad (7)$$

where $\mathbf{E}_v(\mathbf{p})$ is the current error-map value (Sec. 3.3). We then draw $N_{\text{hard}}$ hard pixels from the tile $\tau$ by multinomial sampling based on $\pi_E(\cdot \mid \tau, v)$ *without replacement*.

**Age-Prior Stable Sampling.** Let $\mathcal{H}_{t,v}(\tau)$ be the selected hard pixels in tile $\tau$. We draw age-prior stable pixels from the remaining pixel set, $\tau \setminus \mathcal{H}_{t,v}(\tau)$, as:

$$\mathcal{S}_{t,v}(\tau) = \text{TopK}_{\mathbf{p} \in \tau \setminus \mathcal{H}_{t,v}(\tau)}\big(\mathbf{A}_v(\mathbf{p}), N_{\text{stable}}\big), \quad (8)$$

where $\mathbf{A}_v(\mathbf{p})$ is the age map and $\text{TopK}(\cdot, N_{\text{stable}})$ returns the $N_{\text{stable}}$ pixels with the largest ages.

Finally, the sampled pixels in tile $\tau$ are $\mathcal{P}_{t,v}(\tau) = \mathcal{H}_{t,v}(\tau) \cup \mathcal{S}_{t,v}(\tau)$, and the full sampled set is $\mathcal{P}_{t,v} = \bigcup_\tau \mathcal{P}_{t,v}(\tau)$.

### 3.6. Differentiable Sparse Pixel Rasterization

We output sampled pixels in tile order for GPU tile-parallel execution and perform differentiable rasterization on $\mathcal{P}_{t,v}$, to obtain rendered colors $\hat{\mathbf{I}}_{t,v}(\mathbf{p})$ and Gaussian gradients.

As illustrated in Fig. 6, the forward pass maintains valid sampled tiles and computes tile-wise pixel offsets via prefixsum, enabling each CUDA block (tile) to efficiently access

sampled pixel indices and rasterize only sparse pixels. For the backward pass, following the splat-parallel strategy of Taming-3DGS (Mallick et al., 2024), we retrieve sparse pixels within each tile through tile-pixel indexing and propagate gradients via warp shuffle only over sampled pixels, avoiding dense rasterization replay while preserving efficient gradient accumulation.

### 3.7. Sparse Supervision with Tile-Level NCC

Sparse supervision can be unreliable when relying only on pixel-wise losses (only $\ell_1$ loss in Tab. 4 and w/o S-NCC in Fig. 11). To better preserve local structures, we introduce a tile-level sparse NCC regularization on sampled pixels.

For a tile $\tau$ in view $v$, we compute a sparse Normalized Cross-Correlation (NCC) similarity over the sampled pixels $\mathcal{P}_{t,v}(\tau)$ and define the tile-wise structure loss:

$$\mathcal{L}^\tau_{\text{S-NCC}} = 1 - \text{NCC}\big(\hat{\mathbf{I}}_{t,v}(\mathcal{P}_{t,v}(\tau)), \mathbf{I}_v(\mathcal{P}_{t,v}(\tau))\big). \quad (9)$$

**Gradient-Balanced Weighting.** Instead of using a fixed NCC weight, we balance the loss by the ratio of the average gradient norms of the $\ell_1$ and NCC terms within each tile:

$$\lambda^\tau_{\text{ncc}} = \text{clip}\left(\frac{\bar{g}^\tau_{\ell_1}}{\bar{g}^\tau_{\text{S-NCC}} + \epsilon}, 0, 1\right),$$

$$\bar{g}^\tau_k = \mathbb{E}_{\mathbf{p} \in \mathcal{P}_{t,v}(\tau)}\Big[\|\nabla_{\hat{\mathbf{I}}(\mathbf{p})} \mathcal{L}^\tau_k(\mathbf{p})\|_1\Big], \quad k \in \{\ell_1, \text{S-NCC}\}. \quad (10)$$

In practice, this is computed in a CUDA kernel, where each tile corresponds to one thread block, and $\bar{g}^\tau_k$ is obtained via block-level gradient reduction of pixel gradient norm.

**Sparse Pixel Supervision.** Our sparse pixel supervision is

*Table 1.* **Quantitative comparisons with existing improved 3DGS optimization methods.** TurboGS completes training **within 80 seconds** while achieving competitive rendering quality compared with prior methods. Best, second-best, and third-best results are highlighted by  best score ,  second best score , and  third best score , respectively. Time is reported in seconds.

| Method | \multicolumn{6}{c}{Mip-NeRF 360 (Barron et al., 2022)} | \multicolumn{6}{c}{Tanks & Temples (Knapitsch et al., 2017)} | \multicolumn{6}{c}{Deep Blending (Hedman et al., 2018)} |
| --- | --- | --- | --- | --- | --- | --- | --- | --- | --- | --- | --- | --- | --- | --- | --- | --- | --- | --- |
| | Time↓ | PSNR↑ | SSIM↑ | LPIPS↓ | $N_{GS}$↓ | FPS↑ | Time↓ | PSNR↑ | SSIM↑ | LPIPS↓ | $N_{GS}$↓ | FPS↑ | Time↓ | PSNR↑ | SSIM↑ | LPIPS↓ | $N_{GS}$↓ | FPS↑ |
| 3DGS | 891 | 27.55 | 0.816 | 0.215 | 2.73M | 199 | 583 | 23.84 | 0.853 | 0.169 | 1.58M | 252 | 921 | 29.85 | 0.907 | 0.238 | 2.48M | 208 |
| Taming-3DGS | 148 | 27.26 | 0.795 | 0.259 | 0.67M | 413 | 97 | 23.77 | 0.836 | 0.211 | 0.32M | 612 | 100 | 29.92 | 0.903 | 0.271 | 0.29M | 601 |
| Mini-Splatting | 674 | 27.32 | 0.822 | 0.217 | 0.49M | 279 | 479 | 23.24 | 0.836 | 0.202 | 0.20M | 405 | 586 | 29.95 | 0.907 | 0.254 | 0.35M | 364 |
| Speedy-Splat | 533 | 26.95 | 0.786 | 0.288 | 0.32M | 983 | 292 | 23.41 | 0.820 | 0.239 | 0.19M | 1138 | 497 | 29.55 | 0.903 | 0.268 | 0.25M | 1121 |
| 3DGS-LM | 622 | 27.41 | 0.814 | 0.220 | 3.35M | 218 | 369 | 23.57 | 0.844 | 0.185 | 1.80M | 285 | 572 | 29.58 | 0.906 | 0.246 | 2.89M | 218 |
| DashGaussian | 169 | 27.69 | 0.817 | 0.220 | 2.24M | 259 | 141 | 24.04 | 0.851 | 0.181 | 1.20M | 334 | 112 | 30.12 | 0.907 | 0.248 | 1.95M | 289 |
| FastGS | 111 | 27.51 | 0.805 | 0.261 | 0.38M | 1002 | 91 | 24.06 | 0.842 | 0.209 | 0.24M | 1087 | 81 | 30.06 | 0.905 | 0.267 | 0.22M | 1144 |
| ConeGS | 778 | 27.74 | 0.823 | 0.202 | 0.87M | 427 | 749 | 23.88 | 0.853 | 0.160 | 0.55M | 564 | 805 | 30.28 | 0.909 | 0.238 | 0.52M | 655 |
| **TurboGS (Ours)** | 77 | 27.57 | 0.794 | 0.256 | 0.64M | 153 | 73 | 23.79 | 0.832 | 0.200 | 0.58M | 170 | 62 | 30.00 | 0.900 | 0.277 | 0.49M | 231 |
| **TurboGS-Big (Ours)** | 168 | 27.99 | 0.815 | 0.224 | 1.34M | 134 | 146 | 24.14 | 0.841 | 0.181 | 0.91M | 151 | 132 | 30.26 | 0.909 | 0.259 | 0.69M | 216 |

formulated only by the sampled pixels for each tile:

$$\mathcal{L} = \mathbb{E}_{v \in \mathcal{V}_t, \, \mathbf{p} \in \mathcal{P}_{t,v}} \left[ \ell_1(\mathbf{p}) \right] + \mathbb{E}_{v \in \mathcal{V}_t, \, \tau} \left[ \lambda_{\text{ncc}}^{\tau} \mathcal{L}_{\text{S-NCC}}^{\tau} \right]. \quad (11)$$

In practice, loss evaluation and per-tile gradient statistics are efficiently fused in an independent CUDA kernel for fast execution with low overhead (see runtime in Fig. 7).

### 3.8. Error-Driven Density Control

Besides sparse supervision, TurboGS further adapts the Gaussian density using multi-view errors $e_{t,v}(\mathbf{p})$ to guide densification and pruning. When a Gaussian $g_i$ contributes to the pixel $\mathbf{p}$ with compositing weight $w_{t,i}(\mathbf{p}) = T_{t,i}(\mathbf{p})\,\alpha_{t,i}(\mathbf{p})$, we accumulate per-Gaussian statistics:

$$\mathcal{E}_i^{(t)} = \sum_{v,\mathbf{p}} e_{t,v}(\mathbf{p})\, w_{t,i}(\mathbf{p}), \quad \mathcal{W}_i^{(t)} = \sum_{v,\mathbf{p}} w_{t,i}(\mathbf{p}). \quad (12)$$

Similarly, we accumulate a distance term $\mathcal{D}_i^{(t)}$ by replacing $e_{t,v}(\mathbf{p})$ with a pixel-splat distance term $D_{t,i}(\mathbf{p})$ (*i.e.,* the conic Mahalanobis distance computed from the offset between 2D Gaussian projection and the pixel coordinate).

We normalize the accumulated statistics by visibility: $e_i^{(t)} = \mathcal{E}_i^{(t)}/(\mathcal{W}_i^{(t)} + \epsilon)$ and $d_i^{(t)} = \mathcal{D}_i^{(t)}/(\mathcal{W}_i^{(t)} + \epsilon)$, and define the Gaussian difficulty score as:

$$s_i^{(t)} = e_i^{(t)} + \lambda_{\text{eff}}\, d_i^{(t)}, \qquad \lambda_{\text{eff}} = \lambda_{\text{base}}(1 - \rho_t), \quad (13)$$

where $\rho_t \in [0, 1)$ denotes the training progress. Normalization by $\mathcal{W}_i^{(t)}$ removes bias toward frequently visible Gaussians, while the distance term emphasizes boundary regions that often indicate under-fitting.

Here, different from the density control in FastGS (Ren et al., 2026), which relies on post-hoc dense error maps for densification and pruning, TurboGS uses Gaussian scores $s_i^{(t)}$ and $e_i^{(t)}$ obtained from online sparse pixel statistics, along with their percentile thresholds for adaptive density control, reducing computational complexity for faster training.

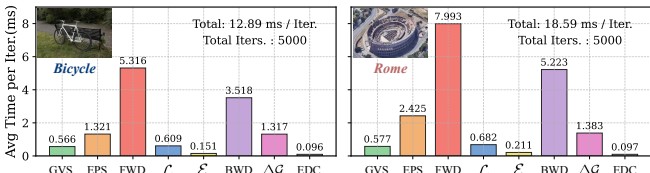

*Figure 7.* **Per-step runtime breakdown of TurboGS.** We report the average execution time (ms) of each optimization step on the *Bicycle* and large-scale *Rome* scenes. Step abbreviations denote the corresponding operations in Alg. 1 (Appendix).

### 3.9. Moment-Damped LM Solver for Sparse Training

Sparse supervision alters the optimization landscape and can amplify gradient variance across views and tiles. To improve convergence while keeping the update cost lightweight, TurboGS adopts a moment-damped Levenberg-Marquardt (LM) solver implemented as per-Gaussian CUDA kernels.

At each iteration, we form a small local equation for each Gaussian $g$ and solve it independently. Let $\mathbf{m}_g$ and $\mathbf{v}_g$ denote the Adam first and second moments for a parameter block, and let $\mathbf{J}_g$ be the local Jacobian estimated from the accumulated gradients. We compute an LM-style update:

$$\left(\mathbf{J}_g^{\top} \mathbf{J}_g + \text{diag}(\sqrt{\mathbf{v}_g} + \epsilon)\right) \Delta_g = -\mathbf{m}_g, \quad (14)$$

where the moment $\mathbf{v}_g$ serves as an diagonal damping to stabilize sparse gradients. Each local formulation is solved via *Cholesky decomposition*, resulting in a fully parallel and lightweight solver tailored for sparse-pixel training.

## 4. Experiments

We evaluate TurboGS through comparisons with representative improved 3DGS optimization methods and ablations of key components. To analyze computational efficiency, we further report the runtime breakdown of each optimization step in Fig. 7. Experimental details (GPU and training settings) are provided in Sec. B in the appendix.

**Datasets and Metrics.** We conduct experiments on three real-world datasets (*i.e.,* Mip-NeRF 360 (Barron et al.,

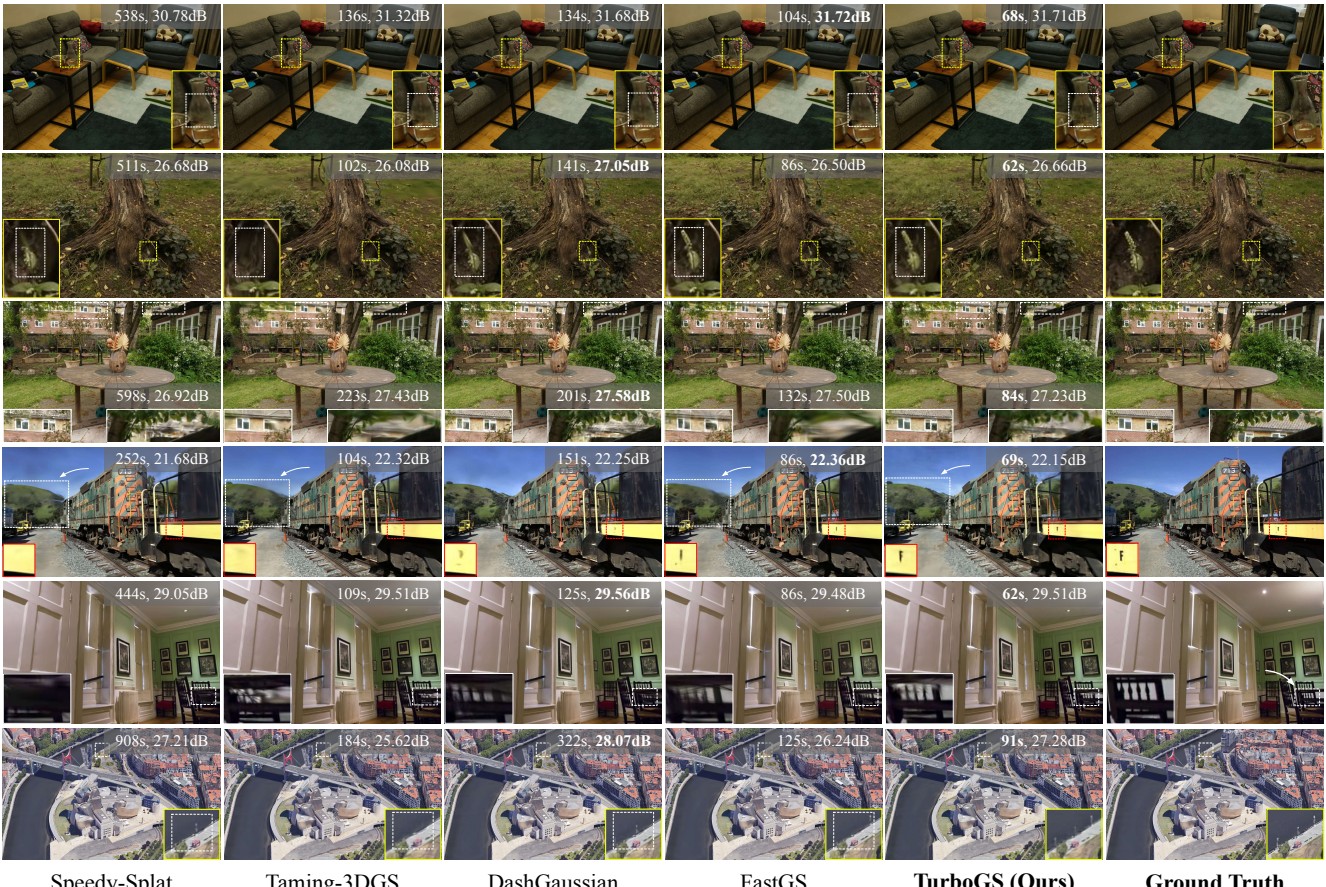

*Figure 8.* **Qualitative comparisons with fast 3DGS optimization methods.** Results are shown on Mip-NeRF 360 (Barron et al., 2022) (*Room*, *Stump* and *Garden* in the first three rows), Tanks & Temples (Knapitsch et al., 2017) (*Train* in the 4th row), Deep Blending (Hedman et al., 2018) (*Drjohnson* in the 5th row), and BungeeNeRF (Xiangli et al., 2022) (*Bilbao* in the last row). TurboGS better preserves some fine structures and details while achieving the **fastest training speed within 100s** among the compared methods.

*Table 2.* **Quantitative comparisons with existing fast 3DGS optimization methods.** Time is reported in seconds.

| Method | BungeeNeRF (*Xiangli et al., 2022*) | | | | | |
|---|---|---|---|---|---|---|
| | Time↓ | PSNR↑ | SSIM↑ | LPIPS↓ | N$_{GS}$ ↓ | FPS↑ |
| Taming-3DGS | 222 | 24.54 | 0.792 | 0.279 | 0.62M | 243 |
| DashGaussian | 332 | 26.55 | **0.870** | **0.172** | 3.05M | 133 |
| FastGS | 136 | 24.97 | 0.812 | 0.256 | **0.61M** | **700** |
| FastGS-Big | 227 | 25.76 | 0.853 | 0.194 | 1.38M | 496 |
| **TurboGS (Ours)** | **101** | 25.69 | 0.808 | 0.252 | 0.84M | 136 |
| **TurboGS-Big** | 230 | **26.68** | 0.863 | 0.180 | 2.25M | 90 |

2022), Tanks & Temples (Knapitsch et al., 2017), and Deep Blending (Hedman et al., 2018)), one large-scale dataset from *Google Earth* imagery used in BungeeNeRF (Xiangli et al., 2022), as well as validations on the OMMO (Lu et al., 2023) dataset and the 4K sub-regions of Rubble (Turki et al., 2022) dataset. Novel view quality is evaluated using average PSNR, SSIM (Wang et al., 2004), and LPIPS (Zhang et al., 2018). Training efficiency and compactness are assessed by total training time (Time, in seconds), the final number of Gaussians (N$_{GS}$), and rendering speed (FPS).

## 4.1. Comparison with 3DGS Optimization Methods

**Baselines.** We compare TurboGS with representative 3DGS optimization methods that target acceleration and primitive control, including Taming-3DGS (Mallick et al., 2024), Speedy-Splat (Hanson et al., 2025a), Mini-Splatting (Fang & Wang, 2024), FastGS (Ren et al., 2026), ConeGS (Baranowski et al., 2026), DashGaussian (Chen et al., 2025), and 3DGS-LM (Höllein et al., 2025), together with vanilla 3DGS (Kerbl et al., 2023) as a reference.

**Quantitative Results.** Tab. 1 and Tab. 2 report quantitative comparisons. TurboGS consistently achieves the fastest training speed across datasets, completing optimization within 80 seconds on three real-world benchmarks, corresponding to a $10\times \sim 14\times$ speedup over vanilla 3DGS and a $1.2\times \sim 2\times$ speedup over recent fast methods. Despite the reduced training cost, TurboGS maintains competitive rendering quality, while our TurboGS-Big (with more iterations; see Sec. B) further achieves the best or near-best PSNR on multiple benchmarks. We attribute this to the error-guided sparse-pixel training scheme, which focuses

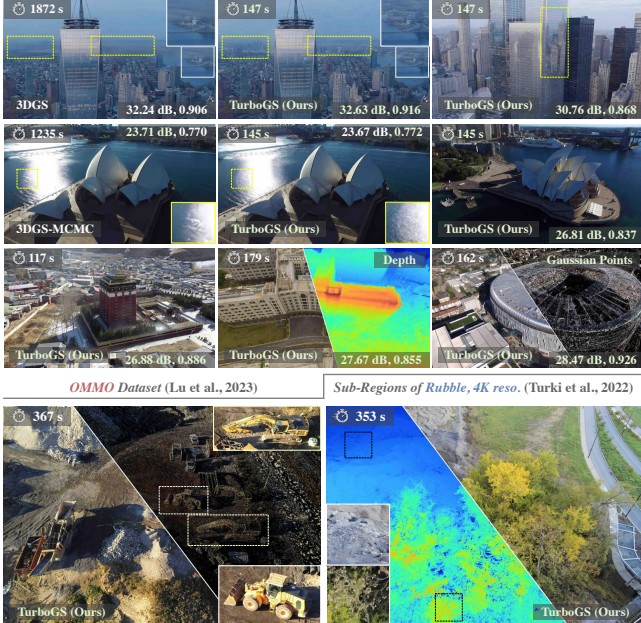

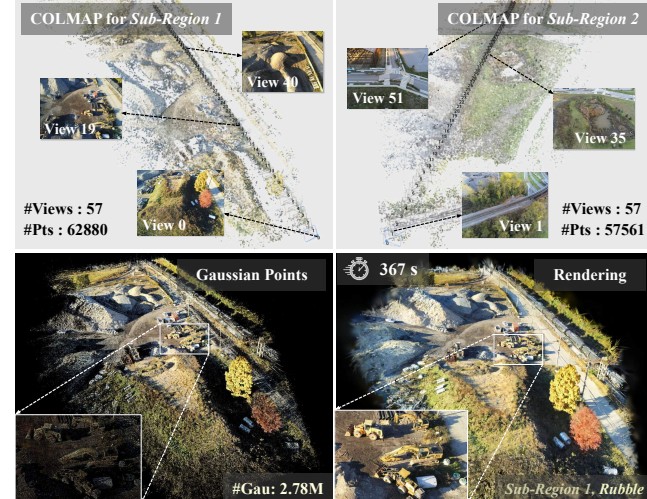

*Figure 10.* **4K Rubble sub-region reconstruction.** Independent COLMAP initialization (cameras and sparse points), and our efficient 3DGS optimization with high-quality rendering results.

*Figure 9.* **Large-scale and high-resolution scene reconstruction.** Results on the large-scale OMMO scene (Lu et al., 2023) and representative 4K Rubble (Turki et al., 2022) sub-regions.

*Table 3.* **Quantitative comparisons on the 4K sub-region Rubble (Turki et al., 2022) dataset.** Time is reported in seconds.

| Method | 4K Sub-Regions of Rubble (Turki et al., 2022) | | | | | |
| | Time↓ | PSNR↑ | SSIM↑ | LPIPS↓ | $N_{GS}$ ↓ | FPS↑ |
|---|---|---|---|---|---|---|
| Taming-3DGS | 898 | 25.45 | 0.733 | 0.400 | **0.94M** | 51 |
| FastGS | 614 | 26.16 | **0.788** | 0.309 | 2.90M | **125** |
| **TurboGS (Ours)** | 360 | **26.32** | 0.771 | 0.284 | 2.73M | 51 |

computation on difficult pixels. Although SSIM and LPIPS are slightly lower in some cases, likely because our pixel-wise training aligns more closely with PSNR, TurboGS achieves a favorable trade-off between speed, quality, and model compactness. Unlike 3DGS-LM, which requires over 32 GB GPU memory, our hybrid optimizer incurs substantially lower computational and memory overhead.

**Qualitative Results.** Fig. 8 presents qualitative comparisons across datasets. Despite the shorter training time, TurboGS preserves some fine structures and local details comparable to or better than prior fast optimization methods. This mainly benefits from error-guided sparse sampling and density control, which help stabilize training in challenging regions. More results are provided in the appendix.

**Large-Scale and High-Resolution Scenes.** Fig. 9 demonstrates the effectiveness of TurboGS on the large-scale OMMO scenes (Lu et al., 2023), showing both training speedup and improved local details compared with vanilla 3DGS and 3DGS-MCMC(Kheradmand et al., 2024). Fig. 10 and Tab. 3 further validate the advantage of TurboGS for high-resolution (*e.g.,* 4K) training, mainly benefiting from

*Table 4.* **Ablation studies of TurboGS.** Experiments are conducted on Mip-NeRF 360 (Barron et al., 2022) dataset with vanilla 3DGS (Kerbl et al., 2023) as the baseline. We progressively add core components and replace key modules to isolate their effects.

| Ablated Method | Time↓ | PSNR↑ | SSIM↑ | LPIPS↓ | $N_{GS}$ ↓ |
|---|---|---|---|---|---|
| 3DGS | 891 | 27.55 | **0.816** | **0.215** | 2.73M |
| + Err-SPS. | 85 | 26.58 | 0.737 | 0.331 | 0.76M |
| + Geo-VS. | 89 | 26.73 | 0.743 | 0.321 | 1.05M |
| + Sparse-NCC. | 95 | 27.14 | 0.773 | 0.279 | 0.98M |
| + Err-DP. **(Full)** | 77 | **27.57** | 0.794 | 0.256 | 0.64M |
| Err-SPS. → RPS. | **75** | 27.10 | 0.780 | 0.276 | 0.56M |
| Geo-VS. → RVS. | 76 | 27.26 | 0.780 | 0.274 | **0.47M** |
| Err-DP. → FastGS | 91 | 27.55 | 0.801 | 0.258 | 0.60M |
| $\ell_1$ loss (only) | 92 | 27.09 | 0.761 | 0.302 | 0.76M |
| S-NCC. → S-SSIM. | 86 | 27.21 | 0.773 | 0.287 | 0.74M |

sparse pixel sampling that reduces computation proportionally to image resolution. For the 4K Rubble benchmark, we extract three representative 4K sub-region image sets and independently run COLMAP for fair comparison. For higher-resolution scenes at extreme view scales, storing error/age maps may introduce additional overhead, which could be alleviated by scalable strategies such as multi-GPU partitioning or periodic GPU–CPU memory offloading.

### 4.2. Ablation Study

We conduct ablation studies on the Mip-NeRF 360 (Barron et al., 2022) dataset with vanilla 3DGS (Kerbl et al., 2023) as the baseline to analyze the contribution of each component in TurboGS. In particular, our ablation design jointly considers optimization efficiency and rendering quality, aiming to analyze how different components contribute to a *balanced trade-off between training speed and visual fidelity*. Quantitative results are reported in Tab. 4 and Tab. 5, with qualitative comparisons shown in Fig. 11.

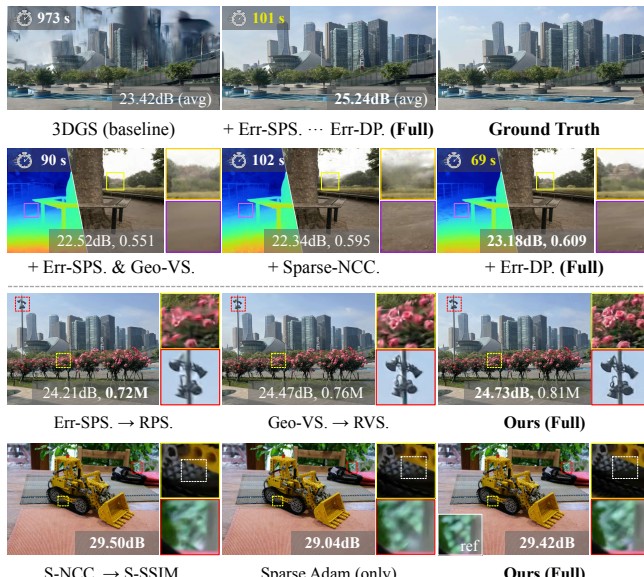

*Figure 11.* **Qualitative ablation results.** Our full model retains finer details than ablated variants while reducing training time. Results are shown on *Park* (Wu et al., 2023) dataset (first and third rows) and Mip-NeRF 360 dataset (second and 4th rows).

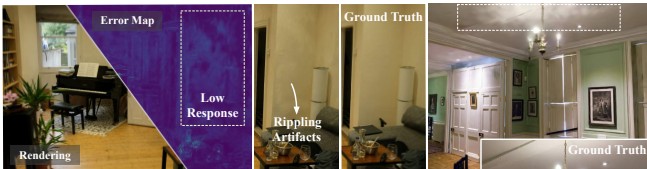

*Figure 12.* **Limitations of TurboGS.** Examples of subtle rippling artifacts in low-frequency regions despite low error-map responses.

**Effect of Adding Key Components.** As shown in the first five rows of Tab. 4, progressively adding our components substantially reduces training time while improving rendering quality. Error-guided sparse pixel sampling ("Err-SPS.") provides most of the acceleration, reducing training time from 891 to 85 seconds and $N_{GS}$ from 2.73M to 0.76M. However, relying only on sparse-pixel $\ell_1$ supervision degrades reconstruction quality. Geometry-aware view sampling ("Geo-VS.") and sparse NCC regularization gradually restore rendering quality and enhance local high-frequency details. After introducing error-driven density control ("Err-DP."), the full TurboGS model achieves the best trade-off, completing training in 77 seconds while improving PSNR by +0.02 dB over the baseline, and reducing $N_{GS}$ to 0.64M. As shown in the first two rows of Fig. 11, the full model removes distant background artifacts and progressively improves fine structural details with faster convergence.

**Sampling Strategies.** Replacing "Err-SPS." with random pixel sampling ("RPS."), and "Geo-VS." with random view sampling ("RVS."), both lead to consistent drops in PSNR and perceptual metrics (the sixth and seventh rows in Tab. 4). Fig. 11 (the third row) also shows that random sampling fails to preserve high-frequency details, with "RPS." causing

*Table 5.* **Optimizer ablation in TurboGS.** Results on Mip-NeRF 360 (Barron et al., 2022), comparing our Moment-LM optimizer against the sparse Adam-only version under identical settings.

| Optimizer | Time↓ | PSNR↑ | SSIM↑ | LPIPS↓ | $N_{GS}$ ↓ |
|---|---|---|---|---|---|
| Sparse Adam (only) | 80 | 27.52 | 0.793 | 0.261 | **0.64M** |
| Moment-LM (**Ours**) | **77** | **27.57** | **0.794** | **0.256** | **0.64M** |

more severe degradation since error-dominant pixels are less likely to be prioritized, resulting in less informative supervision. These results highlight the importance of our pixel-and-view sampling under sparse supervision.

**Gaussian Density Control Strategy.** Replacing our density control ("Err-DP.") with the FastGS-style strategy increases training time and slightly reduces PSNR ( 8th row in Tab. 4), likely due to its reliance on post-hoc full-image error accumulation, which is less efficient in our online sparse setting.

**Objective Functions.** Using only the $\ell_1$ loss degrades reconstruction quality. Replacing sparse NCC ("S-NCC.") with sparse SSIM ("S-SSIM.") slightly increases training time and sometimes yields less sharp local structures (the 4th row in Fig. 11), suggesting that NCC offers more effective structural supervision under sparse training.

**Optimization Strategy.** As shown in Tab. 5, replacing our moment-damped LM with sparse Adam yields slightly lower PSNR/SSIM and higher LPIPS under similar training time. Qualitatively, the Adam-only variant produces less stable high-frequency details (the 4th row in Fig. 11), suggesting that our hybrid optimizer improves stability and reconstruction quality under sparse supervision.

## 5. Conclusion

In this paper, we have proposed TurboGS, an error-guided sparse optimization framework for accelerating 3DGS. TurboGS adaptively reallocates computation from well-reconstructed regions to informative pixels by incorporating tile-wise sparse pixel sampling with persistent error and age maps, geometry-aware view sampling, structure-aware sparse supervision, error-driven density control, and a hybrid moment-damped optimizer. Extensive experiments demonstrate that TurboGS achieves rapid convergence with competitive rendering quality across diverse scenes.

Despite these advantages, TurboGS still has limitations. As shown in Fig. 12, sparse supervision may occasionally introduce subtle rippling artifacts in low-frequency regions with weak texture variations (*e.g.*, walls or ceilings), even when the error-map response is already low. A possible remedy is to incorporate adaptive dense supervision for tiles in such regions to better regularize structural consistency. Moreover, extending TurboGS to dynamic scenes further requires spatiotemporal sampling and cross-frame error aggregation, which we consider an interesting future work.

## Acknowledgements

We thank all the anonymous reviewers for their professional and constructive comments. Weiwei Xu is partially supported by the National Key Research and Development Program of China (No. 2024YFE0216600), and the NSFC grant (No. 62421003). Zheng Dong is supported by the grant of Zhejiang Provincial Natural Science Foundation, China (No. LQN26F020035).

## Impact Statement

This paper focuses on accelerating 3DGS to reduce training overhead while improving the trade-off between computational efficiency and novel view synthesis quality. Although our work may have broader societal implications, we do not foresee any requiring specific discussion at this stage.

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

# Appendix

## A. TurboGS Optimization Procedure

**Algorithm 1 : TurboGS** Training Framework. Steps marked in blue are implemented as custom **CUDA** kernels / operators / buffers; Gray denotes lightweight **CPU** bookkeeping / value.

---

**Input** : Images & cameras $(\mathcal{I}, \mathcal{C})$, initial Gaussians $\mathcal{G}$, total training iterations $T$, sampling view budget range $[K_{\min}, K_{\max}]$, other hyper-params $\Theta$

**Output** : Optimized Gaussians $\mathcal{G}$

**Initialization :** $\bar{\mathcal{E}}_{\text{init}}, \mathcal{E} \leftarrow$ `Initialize_error_maps`$(\mathcal{G}, \mathcal{C}, \mathcal{I})$     `// per-pixel error buffers (GPU)`

**for** $t \leftarrow 0$ **to** $T - 1$ **do**

   $\rho \leftarrow \min(1, t/\tau_{\text{until}})$                                              `// training progress`

   **1) Sampling Parameters :**         `// non-linear query view budget & pixel sampling rate`
   $(K_t, r_t) \leftarrow$ `query_sampling_schedule`$(K_{\max}, K_{\min}, \bar{\mathcal{E}}_{\text{init}}, \Theta, t)$

   **2) View Selection (GVS) :**       `// anchor-by-error, then sampling-by pose/dir distance`
   $\mathcal{V}_t \leftarrow$ `geometry_aware_view_sampling`$(K_t, \rho)$

   **3) Pixel Sampling (EPS) :**             `// tile-wise error-guided sparse pixel sampling`
   $\mathcal{P}_t \leftarrow$ `tile_wise_pixel_sampling`$(\mathcal{V}_t, r_t, \Theta)$

   **4) Gather GT Color & Error :**           `// colors and current errors on sampled pixels`
   $(\mathbf{c}^*, \mathbf{e}^*) \leftarrow$ `get_sampling_pixel_gt_err`$(\mathcal{V}_t, \mathcal{P}_t, \mathcal{I}, \mathcal{E})$

   **5) Sparse Forward Rendering (FWD) :** $\mathbf{o}, \mathbf{d} \leftarrow$ `rasterize_forward_pixels`$(\mathcal{V}_t, \mathcal{P}_t, \mathbf{e}^*)$

   **6) Sparse Loss & Grads. ($\mathcal{L}$) :** $(\nabla\mathbf{c}, \nabla\mathbf{d}, \mathcal{L}_{\text{pix}}) \leftarrow$ `calculate_l1_sncc_grad_loss`$(\mathbf{o}, \mathbf{c}^*, \mathbf{d}, \mathbf{d}^*(optional))$

   **7) EMA Error-Map Update ($\mathcal{E}$) :** $\mathcal{E} \leftarrow$ `update_err_statistics_EMA`$(\mathcal{E}, \mathcal{L}_{\text{pix}}, \mathcal{V}_t, \mathcal{P}_t)$

   **8) Gaussian Backward & Screen Properties (BWD) :**      `// gradients of Gaussian properties`
   $\nabla\mathcal{G} \leftarrow$ `rasterize_backward_pixels`$(\nabla\mathbf{c}, \nabla\mathbf{d}, \mathcal{G})$
   $(\nabla\mathcal{G}_{screen}, \mathbf{r}) \leftarrow$ `get_gaussian_grad_screen_radii`$(\nabla\mathcal{G}, \mathcal{G})$

   **9) Solve Gaussian Updates ($\Delta\mathcal{G}$) :** $\Delta\mathcal{G} \leftarrow$ `solve_gaussian_delta_properties`$(\nabla\mathcal{G}, t, \rho)$

   **10) Apply Gaussian Update :** $\mathcal{G} \leftarrow$ `update_gaussian_properties`$(\mathcal{G}, \Delta\mathcal{G})$

   **11) Error-Driven Density Control (EDC):**      `// adaptive Gaussian densification & pruning`
     **Back-Projection Pixel Properties to Gaussians :**      `// accumulate (err,dist,`$\alpha$`) per Gaussian`
     $e_{\mathcal{G}}, D_{\mathcal{G}}, \alpha_{\mathcal{G}} \leftarrow$ `rasterize_forward_pixels`$(\mathcal{V}_t, \mathcal{P}_t, \mathcal{L}_{\text{pix}})$
     $\mathcal{G} \leftarrow$ `error_driven_densify_and_prune`$(\mathcal{G}, \{\nabla\mathcal{G}_{screen}, \mathbf{r}\}, \{e_{\mathcal{G}}, D_{\mathcal{G}}, \alpha_{\mathcal{G}}\}, t, \Theta)$

**return** $\mathcal{G}$

---

Alg. 1 outlines the complete optimization pipeline of TurboGS, illustrating how error-guided sampling and sparse supervision are integrated into the 3DGS training loop. We highlight the input-output variable relationships between our *sparse sampling*, *sparse supervision*, *Gaussian property optimization*, and *error-driven density control*. In addition to the core algorithmic details already explained in the main paper, we further clarify the details of several steps:

**Adaptive Scheduling of View Budget and Pixel Sampling.** TurboGS dynamically adjusts both the sampled view budget and tile-wise pixel sampling rate according to reconstruction difficulty and training progress. At iteration $t$, we first compute a normalized error ratio:

$$r_t = \text{clip}\left(\frac{\bar{E}_t}{\bar{E}_{\text{init}} + \epsilon}, 0, 1\right), \qquad \phi(r_t) = \sin\left(\tfrac{\pi}{2}\sqrt{r_t}\right), \tag{15}$$

where $\bar{E}_t$ and $\bar{E}_{\text{init}}$ denote the current and initial mean view errors. The sinusoidal modulation is motivated by the observation that reconstruction error drops rapidly in early training and then remains within a narrow range for most iterations (see Fig. 2 and Fig. 5). It amplifies sampling variation in this long low-error regime, improving sensitivity to subtle reconstruction differences.

We further define a progress coefficient $\rho_t = 1 - \text{clip}(t/\tau_{\text{until}}, 0, 1)$ to gradually anneal sampling after densification. The tile-wise pixel sampling rate and sampled view number are jointly scheduled as:

$$r_{\text{tile}}^{(t)} = r_{\min} + (r_{\max} - r_{\min})\big(1 - \rho_t\phi(r_t)\big), \qquad K_t = \text{round}\Big(K_{\min} + (K_{\max} - K_{\min})e^{-\alpha(1 - \rho_t\phi(r_t))}\Big), \tag{16}$$

where $r_{\min} = \lambda/K_{\max}$, $r_{\max} = \lambda/K_{\min}$, $\lambda$ denotes the tile sampling scale, and $\alpha$ controls the transition speed. This design allocates more views and denser pixel sampling during difficult stages, while progressively reducing computation and shifting from global to more local view exploration as optimization converges.

**Gradient Computation under Sparse Supervision.** Under sparse pixel sampling, TurboGS computes gradients by combining a pixel-wise Charbonnier $\ell_1$ term and a tile-level sparse NCC term. For a sampled pixel with prediction $x$ and ground truth $y$, the gradients are:

$$\frac{\partial \ell_1}{\partial x} = \frac{x - y}{\sqrt{(x - y)^2 + \epsilon^2}}, \qquad \frac{\partial \ell_{\text{NCC}}}{\partial x} = -\left( \frac{y - \mu_y}{\sqrt{\sigma_x^2 \sigma_y^2 + \epsilon}} - \text{NCC} \frac{x - \mu_x}{\sigma_x^2 + \epsilon} \right), \qquad (17)$$

where $\mu_x, \mu_y$ and $\sigma_x^2, \sigma_y^2$ denote the tile-wise mean and variance computed from sampled pixels.

To stabilize sparse optimization, we adaptively balance the two terms using tile-level gradient statistics:

$$\lambda_{\text{ncc}}^\tau = \text{clip}\left( \frac{\mathbb{E}[|\nabla \ell_1|]}{\mathbb{E}[|\nabla \ell_{\text{NCC}}|] + \epsilon}, 0, 1 \right), \qquad \frac{\partial \mathcal{L}}{\partial x} = \frac{\partial \ell_1}{\partial x} - \lambda_{\text{ncc}}^\tau \frac{\partial \ell_{\text{NCC}}}{\partial x}. \qquad (18)$$

**Details of Our Moment-Damped LM Optimizer.** TurboGS adopts a hybrid optimization strategy that combines lightweight Levenberg–Marquardt (LM) updates with Adam-style moment damping. For each Gaussian, we partition the parameters into two compact blocks: geometry parameters $\mathbf{g} \in \mathbb{R}^{10}$ (position, scale, rotation) and base appearance parameters $\mathbf{a} \in \mathbb{R}^4$ (RGB DC and opacity), which are solved independently through small LM systems:

$$(\mathbf{J}^\top \mathbf{J} + \text{diag}(\sqrt{\mathbf{v}} + \epsilon))\Delta = -\mathbf{m}, \qquad \Delta \boldsymbol{\theta}_{\text{SH}} = -\eta \frac{\hat{\mathbf{m}}}{\sqrt{\hat{\mathbf{v}}} + \epsilon}, \qquad (19)$$

where $\mathbf{m}$ and $\mathbf{v}$ denote Adam first and second moments, and $\mathbf{v}$ serves as an adaptive diagonal damping term. Each LM system is efficiently solved via Cholesky decomposition.

Higher-order spherical harmonic (SH) coefficients are updated separately using Adam to avoid enlarging the LM system size. Ignoring SH cross-parameter correlations is a practical trade-off between accuracy and efficiency, since joint optimization would substantially increase Jacobian size and memory overhead. Empirically, this hybrid strategy improves stability under sparse gradients without noticeable convergence degradation. In practice, LM-based updates and SH optimization are executed in two dedicated CUDA kernels for efficient parallelism.

**Convergence Stability under Sparse Optimization.** TurboGS adopts a structured sparse optimization scheme whose stochastic gradients remain aligned with the dense objective. Let the dense loss be $\mathcal{L}(\theta) = \sum_{i=1}^N \ell_i(\theta)$ over image pixels $i$. Under sparse sampling set $\mathcal{S}$, the gradient estimator is:

$$\hat{\mathbf{g}} = \frac{1}{|\mathcal{S}|} \sum_{i \in \mathcal{S}} w(i) \nabla \ell_i(\theta), \qquad \mathbb{E}[\hat{\mathbf{g}}] = \sum_{i=1}^N p(i) w(i) \nabla \ell_i(\theta), \qquad (20)$$

where $p(i)$ denotes the sampling probability and $w(i)$ is the importance weight. Under ideal importance sampling with $w(i) = 1/p(i)$, the estimator recovers the dense gradient in expectation, *i.e.*, $\mathbb{E}[\hat{\mathbf{g}}] \propto \nabla \mathcal{L}(\theta)$, providing convergence consistency while reducing computation.

In practice, TurboGS combines error-guided hard sampling and age-aware stable sampling, prioritizing informative pixels while periodically revisiting under-sampled regions to avoid overly localized supervision. Moreover, the moment-damped LM update improves optimization stability by introducing adaptive diagonal damping through the Adam second moment, $\mathbf{J}^\top \mathbf{J} + \text{diag}(\sqrt{\mathbf{v}} + \epsilon)$, which improves local conditioning under sparse gradients. Together, these designs lead to stable empirical convergence in sparse-pixel optimization.

# B. Implementation Details

All experiments are conducted on an NVIDIA RTX 5090 GPU, except for 3DGS-LM (Höllein et al., 2025), which is evaluated on an NVIDIA RTX PRO 6000 GPU due to its memory requirement exceeding 32 GB. The RTX PRO 6000

*Table 6.* **Definition and values of the hyper-parameters $\Theta$ in TurboGS.**

| H.P. | Definition | Value |
|------|-----------|-------|
| $T$ | Total training iterations | 5000 (TurboGS, fast), 8000 (TurboGS-Big) |
| $\tau_{\text{until}}$ | Last iteration for densification and pruning | 3500 (TurboGS, fast), 6000 (TurboGS-Big) |
| $\tau_{\text{from}}$ | First iteration for densification | 200 |
| $K_{\max}$ | Maximum sampled views per iteration | 10 |
| $K_{\min}$ | Minimum sampled views per iteration | 3 |
| $\lambda$ | Tile pixel sampling scale | $(0, 1]$ |
| $\alpha$ | Transition speed of adaptive view scheduling | 3.0 |
| $\Delta_{\text{dens}}$ | Densification interval (iterations) | $100 \sim 350$ |
| $\Delta_{\text{prune}}$ | Final pruning interval (iterations) | 500 |
| $\eta_{\text{topk}}$ | Fraction for top-$k$ densification | $0.5 \sim 0.95$ |
| $\eta_{\text{final}}$ | Fraction retained by final pruning | $0.8 \sim 0.95$ |
| $\tau_{\text{grad}}$ | Gradient threshold for densification | $1.0 \times 10^{-4} \sim 2.0 \times 10^{-4}$ |
| $\tau_{\text{grad}}^{\text{abs}}$ | Absolute-gradient threshold | $2.0 \times 10^{-4} \sim 4.0 \times 10^{-4}$ |
| $\rho_{\text{dense}}$ | Percent dense | $0.001 \sim 0.02$ |
| $r_{\text{hard}}$ | Sampling ratio for difficult pixels | 0.4 |
| $r_{\text{stable}}$ | Sampling ratio for long-untrained pixels | 0.3 |
| $-$ | Remaining ratio for mixed sampling | 0.3 |
| $\beta$ | EMA blending ratio for pixel errors | 0.4 |
| $\lambda_{\text{base}}$ | Base weight of distance-aware Gaussian difficulty | 0.8 |

provides comparable compute throughput (TFLOPS) to the RTX 5090 while offering 96 GB GPU memory. All baselines are evaluated using their official implementations under the original 3DGS setting. TurboGS is implemented in CUDA and PyTorch, with sparse rasterization, loss and gradient evaluation, and optimization executed as custom CUDA kernels. Training hyper-parameters are summarized in Tab. 6.

# C. Ablation on Sampling Hyper-parameters

*Table 7.* **Ablations on adaptive sampling hyper-parameters.** We evaluate the effects of the adaptive view budget ($K_{\max} \to K_{\min}$), hard-pixel sampling ratio ($r_{\text{hard}}$), and error-ratio scheduling function $\phi(r_t)$ on Mip-NeRF 360 (Barron et al., 2022) under the same setup.

| $K$ | Time↓ | PSNR↑ | SSIM↑ | LPIPS↓ | $N_{GS}$ ↓ |
|-----|-------|-------|-------|--------|-----------|
| 1 (fixed) | 81s | 26.50 | 0.768 | 0.282 | 1.24M |
| 3 (fixed) | 78s | 27.23 | 0.787 | 0.263 | 0.74M |
| 10 ⇒ 3 (**Ours**) | **77s** | **27.57** | **0.794** | **0.256** | 0.64M |
| 10 (fixed) | 121 | 27.41 | 0.783 | 0.263 | **0.51M** |

(a) Ablation on adaptive view budget

| $r_{\text{hard}}$ | Time↓ | PSNR↑ | SSIM↑ | LPIPS↓ | $N_{GS}$ ↓ |
|-----|-------|-------|-------|--------|-----------|
| 0.2 | **77s** | 27.29 | 0.785 | 0.266 | **0.59M** |
| 0.4 (**Ours**) | **77s** | **27.57** | **0.794** | **0.256** | 0.64M |
| 0.6 | 82s | 27.33 | 0.787 | 0.262 | 0.65M |
| 0.8 | 80s | 27.41 | 0.785 | 0.264 | 0.60M |

(b) Ablation on hard-pixel sampling ratio

| Schedule | Time↓ | PSNR↑ | SSIM↑ | LPIPS↓ | $N_{GS}$ ↓ |
|----------|-------|-------|-------|--------|-----------|
| *Linear* | 81s | 27.35 | 0.788 | 0.261 | 0.63M |
| *Exponential* | 79s | 27.27 | 0.783 | 0.268 | **0.55M** |
| *Sinusoidal* (Eq. 15, **Ours**) | **77s** | **27.57** | **0.794** | **0.256** | 0.64M |

(c) Ablation on error-ratio scheduling

We further analyze the sensitivity of TurboGS to key sampling hyper-parameters, including the adaptive view budget, hard-pixel sampling ratio ($r_{\text{hard}}$), and error-ratio scheduling function $\phi(r_t)$. Results are reported in Tab. 7.

**Effect of Adaptive View Budget.** As shown in Tab. 7(a), increasing the number of sampled views improves reconstruction quality but incurs higher computation. A very small view budget ($K{=}1$) degrades performance due to insufficient multi-view constraints, while a large fixed budget ($K{=}10$) increases training time without gains. Our adaptive strategy ($10{\to}3$) achieves the best trade-off by gradually shifting from global multi-view consistency to local refinement.

**Effect of Hard-Pixel Sampling Ratio.** Tab. 7(b) shows that a small $r_{\text{hard}}$ under-samples informative regions, whereas a large value reduces sampling diversity and increases cost. We find $r_{\text{hard}}{=}0.4$ provides a balance between speed and quality.

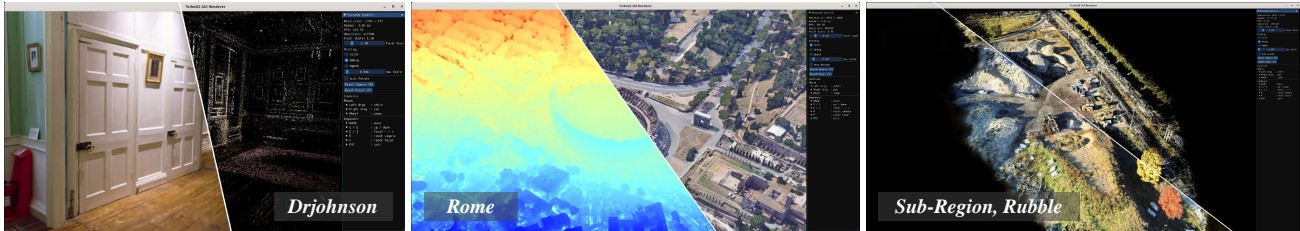

*Figure 13.* **Interactive GUI of TurboGS.** Real-time rendering and free-viewpoint exploration of reconstructed scenes.

**Effect of Error-Ratio Scheduling.** Tab. 7(c) compares different scheduling functions for adaptive sampling. The proposed sinusoidal schedule consistently outperforms linear and exponential variants, likely because it amplifies sampling variation in the long low-error regime, improving sensitivity to subtle reconstruction differences.

## D. Additional Scene-wise Comparisons and Visualizations

We provide additional scene-wise evaluations and visualizations. Tab. 8, Tab. 9, Tab. 10, Tab. 11, and Tab. 12 report detailed scene-wise quantitative comparisons on Mip-NeRF 360, Tanks & Temples, Deep Blending, BungeeNeRF, and 4K Rubble sub-regions, respectively. Fig. 13 presents the interactive GUI of our reconstruction system. Fig. 14 and Fig. 15 visualize intermediate optimization results on the large-scale OMMO dataset and 4K Rubble sub-regions, respectively. In addition, Fig. 16 shows further qualitative comparisons on representative scenes under matched training time.

*Table 8.* **Scene-wise quantitative results on Mip-NeRF 360 (Barron et al., 2022) dataset.** We report the per-scene comparisons of TurboGS and other accelerated or improved 3DGS optimization methods.

| Method | Bicycle | | | | | Flowers | | | | | Garden | | | | |
|---|---|---|---|---|---|---|---|---|---|---|---|---|---|---|---|
| | Time↓ | PSNR↑ | SSIM↑ | LPIPS↓ | N_GS ↓ | FPS↑ | Time↓ | PSNR↑ | SSIM↑ | LPIPS↓ | N_GS ↓ | FPS↑ | Time↓ | PSNR↑ | SSIM↑ | LPIPS↓ | N_GS ↓ | FPS↑ |

| Method | Time↓ | PSNR↑ | SSIM↑ | LPIPS↓ | N_GS↓ | FPS↑ | Time↓ | PSNR↑ | SSIM↑ | LPIPS↓ | N_GS↓ | FPS↑ | Time↓ | PSNR↑ | SSIM↑ | LPIPS↓ | N_GS↓ | FPS↑ |
|---|---|---|---|---|---|---|---|---|---|---|---|---|---|---|---|---|---|---|
| | *Bicycle* | | | | | | *Flowers* | | | | | | *Garden* | | | | | |
| 3DGS | 1069 | 25.27 | 0.767 | 0.208 | 4.91M | 105 | 706 | 21.57 | 0.605 | 0.336 | 2.90M | 207 | 1105 | 27.52 | 0.868 | 0.107 | 4.14M | 138 |
| Taming-3DGS | 139 | 24.82 | 0.717 | 0.294 | 0.81M | 407 | 118 | 21.10 | 0.555 | 0.407 | 0.57M | 526 | 223 | 27.43 | 0.858 | 0.126 | 1.94M | 320 |
| Mini-Splatting | 524 | 25.20 | 0.772 | 0.225 | 0.53M | 241 | 569 | 21.42 | 0.625 | 0.327 | 0.57M | 224 | 566 | 26.84 | 0.847 | 0.150 | 0.56M | 233 |
| Speedy-Splat | 584 | 24.90 | 0.724 | 0.303 | 0.61M | 777 | 470 | 21.39 | 0.579 | 0.395 | 0.36M | 982 | 598 | 26.92 | 0.834 | 0.183 | 0.53M | 843 |
| 3DGS-LM | 727 | 25.21 | 0.764 | 0.219 | 6.06M | 105 | 524 | 21.49 | 0.600 | 0.342 | 3.60M | 196 | 762 | 27.33 | 0.865 | 0.110 | 5.93M | 116 |
| DashGaussian | 225 | 25.38 | 0.771 | 0.213 | 4.26M | 197 | 168 | 21.94 | 0.617 | 0.327 | 2.50M | 302 | 201 | 27.58 | 0.863 | 0.121 | 2.82M | 270 |
| FastGS | 100 | 24.82 | 0.722 | 0.296 | 0.48M | 1057 | 104 | 21.37 | 0.575 | 0.388 | 0.45M | 970 | 132 | 27.50 | 0.847 | 0.157 | 0.66M | 958 |
| ConeGS | 694 | 25.50 | 0.782 | 0.188 | 1.27M | 354 | 646 | 21.55 | 0.625 | 0.303 | 1.00M | 393 | 671 | 27.65 | 0.870 | 0.106 | 1.32M | 362 |
| **TurboGS (Ours)** | 73 | 25.27 | 0.716 | 0.291 | 0.85M | 187 | 73 | 22.06 | 0.580 | 0.387 | 0.78M | 131 | 84 | 27.23 | 0.823 | 0.208 | 0.75M | 166 |
| **TurboGS-Big (Ours)** | 160 | 25.66 | 0.747 | 0.257 | 1.87M | 147 | 180 | 22.61 | 0.630 | 0.327 | 2.36M | 118 | 154 | 27.62 | 0.858 | 0.155 | 1.06M | 148 |
| | *Stump* | | | | | | *Treehill* | | | | | | *Room* | | | | | |
| 3DGS | 898 | 26.66 | 0.771 | 0.216 | 4.29M | 156 | 826 | 22.59 | 0.634 | 0.326 | 3.27M | 175 | 880 | 31.62 | 0.921 | 0.217 | 1.30M | 234 |
| Taming-3DGS | 102 | 26.08 | 0.736 | 0.292 | 0.48M | 568 | 134 | 23.03 | 0.625 | 0.384 | 0.78M | 420 | 136 | 31.32 | 0.909 | 0.249 | 0.23M | 389 |
| Mini-Splatting | 504 | 27.17 | 0.806 | 0.198 | 0.61M | 217 | 564 | 22.72 | 0.654 | 0.314 | 0.57M | 218 | 780 | 31.27 | 0.921 | 0.212 | 0.40M | 440 |
| Speedy-Splat | 511 | 26.68 | 0.770 | 0.259 | 0.51M | 876 | 460 | 22.50 | 0.592 | 0.443 | 0.36M | 1066 | 538 | 30.78 | 0.896 | 0.278 | 0.12M | 1088 |
| 3DGS-LM | 994 | 26.70 | 0.776 | 0.218 | 4.86M | 154 | 519 | 22.55 | 0.634 | 0.332 | 3.80M | 179 | 520 | 31.31 | 0.915 | 0.222 | 1.54M | 283 |
| DashGaussian | 141 | 27.05 | 0.783 | 0.216 | 3.42M | 313 | 90 | 23.06 | 0.640 | 0.319 | 3.20M | 257 | 134 | 31.68 | 0.916 | 0.230 | 1.04M | 245 |
| FastGS | 86 | 26.50 | 0.751 | 0.283 | 0.35M | 1081 | 90 | 22.67 | 0.674 | 0.416 | 0.36M | 1040 | 104 | 31.72 | 0.913 | 0.239 | 0.21M | 1045 |
| ConeGS | 675 | 27.19 | 0.804 | 0.182 | 1.24M | 375 | 604 | 22.65 | 0.605 | 0.351 | 0.76M | 473 | 941 | 32.13 | 0.927 | 0.200 | 0.47M | 524 |
| **TurboGS (Ours)** | 62 | 26.66 | 0.742 | 0.279 | 0.65M | 179 | 69 | 23.18 | 0.609 | 0.392 | 0.69M | 164 | 68 | 31.71 | 0.915 | 0.215 | 0.45M | 155 |
| **TurboGS-Big (Ours)** | 171 | 27.37 | 0.784 | 0.227 | 2.04M | 137 | 114 | 23.22 | 0.614 | 0.368 | 1.17M | 130 | 173 | 32.15 | 0.925 | 0.191 | 0.95M | 137 |
| | *Counter* | | | | | | *Kitchen* | | | | | | *Bonsai* | | | | | |
| 3DGS | 836 | 29.07 | 0.909 | 0.200 | 1.08M | 242 | 1006 | 31.28 | 0.928 | 0.126 | 1.59M | 202 | 692 | 32.35 | 0.943 | 0.203 | 1.07M | 332 |
| Taming-3DGS | 156 | 28.61 | 0.898 | 0.223 | 0.31M | 341 | 179 | 31.15 | 0.922 | 0.141 | 0.48M | 335 | 147 | 31.78 | 0.935 | 0.219 | 0.41M | 412 |
| Mini-Splatting | 896 | 28.61 | 0.905 | 0.198 | 0.41M | 358 | 900 | 31.24 | 0.926 | 0.129 | 0.43M | 272 | 764 | 31.41 | 0.939 | 0.200 | 0.36M | 310 |
| Speedy-Splat | 538 | 28.18 | 0.869 | 0.274 | 0.10M | 1058 | 589 | 30.01 | 0.890 | 0.202 | 0.11M | 1055 | 508 | 31.18 | 0.920 | 0.260 | 0.13M | 1102 |
| 3DGS-LM | 504 | 28.83 | 0.907 | 0.204 | 1.21M | 300 | 596 | 31.30 | 0.928 | 0.127 | 1.82M | 239 | 453 | 31.97 | 0.941 | 0.206 | 1.25M | 386 |
| DashGaussian | 139 | 28.95 | 0.903 | 0.209 | 0.79M | 250 | 197 | 31.44 | 0.921 | 0.141 | 1.26M | 199 | 131 | 32.09 | 0.940 | 0.206 | 0.84M | 295 |
| FastGS | 116 | 29.10 | 0.900 | 0.219 | 0.21M | 988 | 153 | 31.75 | 0.925 | 0.136 | 0.39M | 880 | 118 | 32.15 | 0.937 | 0.212 | 0.28M | 998 |
| ConeGS | 991 | 28.97 | 0.910 | 0.189 | 0.56M | 410 | 828 | 31.68 | 0.934 | 0.117 | 0.74M | 410 | 953 | 32.37 | 0.947 | 0.185 | 0.46M | 540 |
| **TurboGS (Ours)** | 85 | 28.63 | 0.898 | 0.204 | 0.49M | 125 | 92 | 31.55 | 0.923 | 0.131 | 0.59M | 148 | 85 | 31.82 | 0.939 | 0.202 | 0.55M | 122 |
| **TurboGS-Big (Ours)** | 207 | 29.14 | 0.906 | 0.186 | 0.80M | 123 | 191 | 31.81 | 0.926 | 0.127 | 0.73M | 135 | 162 | 32.31 | 0.945 | 0.176 | 1.07M | 131 |

*Table 9.* **Scene-wise quantitative results over the Tanks & Temples (Knapitsch et al., 2017) dataset.**

| Method | Truck | | | | | | Train | | | | | |
|---|---|---|---|---|---|---|---|---|---|---|---|---|
| | Time↓ | PSNR↑ | SSIM↑ | LPIPS↓ | $N_{GS}$ ↓ | FPS↑ | Time↓ | PSNR↑ | SSIM↑ | LPIPS↓ | $N_{GS}$ ↓ | FPS↑ |
| 3DGS | 625 | 25.47 | 0.884 | 0.142 | 2.07M | 231 | 540 | 22.22 | 0.822 | 0.196 | 1.09M | 272 |
| Taming-3DGS | 90 | 25.22 | 0.868 | 0.184 | 0.27M | 659 | 104 | 22.32 | 0.804 | 0.238 | 0.37M | 565 |
| Mini-Splatting | 527 | 25.05 | 0.874 | 0.160 | 0.19M | 329 | 431 | 21.42 | 0.799 | 0.244 | 0.21M | 482 |
| Speedy-Splat | 332 | 25.13 | 0.868 | 0.189 | 0.26M | 1122 | 252 | 21.68 | 0.773 | 0.289 | 0.11M | 1153 |
| 3DGS-LM | 424 | 25.37 | 0.881 | 0.152 | 2.51M | 235 | 314 | 21.77 | 0.807 | 0.218 | 1.09M | 335 |
| DashGaussian | 130 | 25.84 | 0.885 | 0.152 | 1.40M | 359 | 151 | 22.25 | 0.818 | 0.209 | 1.01M | 309 |
| FastGS | 96 | 25.76 | 0.877 | 0.177 | 0.25M | 1134 | 86 | 22.36 | 0.807 | 0.240 | 0.23M | 1040 |
| ConeGS | 740 | 25.75 | 0.890 | 0.116 | 0.65M | 511 | 758 | 22.01 | 0.816 | 0.204 | 0.46M | 617 |
| **TurboGS (Ours)** | 76 | 25.43 | 0.873 | 0.166 | 0.75M | 159 | 69 | 22.15 | 0.791 | 0.234 | 0.41M | 182 |
| **TurboGS-Big (Ours)** | 148 | 25.82 | 0.883 | 0.146 | 1.16M | 144 | 144 | 22.46 | 0.799 | 0.216 | 0.65M | 158 |

*Table 10.* **Scene-wise quantitative results over the Deep Blending (Hedman et al., 2018) dataset.**

| Method | Drjohnson | | | | | | Playroom | | | | | |
|---|---|---|---|---|---|---|---|---|---|---|---|---|
| | Time↓ | PSNR↑ | SSIM↑ | LPIPS↓ | $N_{GS}$ ↓ | FPS↑ | Time↓ | PSNR↑ | SSIM↑ | LPIPS↓ | $N_{GS}$ ↓ | FPS↑ |
| 3DGS | 1046 | 29.45 | 0.905 | 0.236 | 1.84M | 163 | 795 | 30.25 | 0.910 | 0.240 | 3.11M | 252 |
| Taming-3DGS | 109 | 29.51 | 0.903 | 0.267 | 0.40M | 561 | 91 | 30.33 | 0.903 | 0.276 | 0.18M | 640 |
| Mini-Splatting | 632 | 29.40 | 0.904 | 0.257 | 0.38M | 340 | 539 | 30.49 | 0.911 | 0.250 | 0.32M | 387 |
| Speedy-Splat | 444 | 29.05 | 0.900 | 0.267 | 0.31M | 1094 | 549 | 30.05 | 0.907 | 0.270 | 0.19M | 1149 |
| 3DGS-LM | 645 | 28.97 | 0.903 | 0.247 | 3.48M | 172 | 498 | 30.19 | 0.908 | 0.246 | 2.30M | 264 |
| DashGaussian | 125 | 29.56 | 0.905 | 0.247 | 2.53M | 248 | 99 | 30.67 | 0.910 | 0.249 | 1.37M | 329 |
| FastGS | 86 | 29.48 | 0.900 | 0.272 | 0.25M | 1137 | 76 | 30.64 | 0.911 | 0.262 | 0.13M | 1152 |
| ConeGS | 826 | 29.84 | 0.905 | 0.241 | 0.61M | 610 | 783 | 30.72 | 0.913 | 0.235 | 0.44M | 700 |
| **TurboGS (Ours)** | 62 | 29.53 | 0.892 | 0.285 | 0.42M | 273 | 61 | 30.47 | 0.908 | 0.270 | 0.55M | 188 |
| **TurboGS-Big (Ours)** | 139 | 29.78 | 0.904 | 0.260 | 0.67M | 204 | 125 | 30.73 | 0.913 | 0.258 | 0.72M | 227 |

*Table 11.* **Scene-wise quantitative results over the BungeeNeRF (Xiangli et al., 2022) dataset.**

| Method | Amsterdam | | | | | | Bilbao | | | | | | Hollywood | | | | | |
|---|---|---|---|---|---|---|---|---|---|---|---|---|---|---|---|---|---|---|
| | Time↓ | PSNR↑ | SSIM↑ | LPIPS↓ | $N_{GS}$ ↓ | FPS↑ | Time↓ | PSNR↑ | SSIM↑ | LPIPS↓ | $N_{GS}$ ↓ | FPS↑ | Time↓ | PSNR↑ | SSIM↑ | LPIPS↓ | $N_{GS}$ ↓ | FPS↑ |
| Taming-3DGS | 234 | 24.45 | 0.804 | 0.257 | 0.65M | 243 | 184 | 25.62 | 0.825 | 0.246 | 0.44M | 278 | 198 | 23.72 | 0.713 | 0.363 | 0.55M | 248 |
| DashGaussian | 350 | 26.68 | 0.880 | 0.159 | 3.17M | 125 | 322 | 28.07 | 0.900 | 0.135 | 2.75M | 126 | 347 | 25.83 | 0.843 | 0.192 | 3.45M | 143 |
| FastGS | 129 | 24.82 | 0.816 | 0.247 | 0.56M | 628 | 125 | 26.24 | 0.846 | 0.220 | 0.55M | 766 | 143 | 24.22 | 0.754 | 0.317 | 0.72M | 743 |
| FastGS-Big | 212 | 25.81 | 0.857 | 0.191 | 1.30M | 509 | 207 | 27.11 | 0.875 | 0.170 | 1.28M | 549 | 232 | 24.68 | 0.793 | 0.256 | 1.41M | 476 |
| **TurboGS (Ours)** | 96 | 25.40 | 0.803 | 0.260 | 0.69M | 150 | 91 | 27.28 | 0.843 | 0.219 | 0.76M | 145 | 105 | 24.75 | 0.745 | 0.314 | 0.93M | 132 |
| **TurboGS-Big (Ours)** | 236 | 26.57 | 0.863 | 0.184 | 1.86M | 99 | 227 | 27.94 | 0.880 | 0.168 | 1.86M | 95 | 258 | 25.78 | 0.817 | 0.221 | 2.51M | 85 |

| Method | Pompidou | | | | | | Rome | | | | | |
|---|---|---|---|---|---|---|---|---|---|---|---|---|
| | Time↓ | PSNR↑ | SSIM↑ | LPIPS↓ | $N_{GS}$ ↓ | FPS↑ | Time↓ | PSNR↑ | SSIM↑ | LPIPS↓ | $N_{GS}$ ↓ | FPS↑ |
| Taming-3DGS | 230 | 24.04 | 0.830 | 0.234 | 0.68M | 271 | 224 | 24.11 | 0.794 | 0.278 | 0.62M | 242 |
| DashGaussian | 409 | 26.44 | 0.903 | 0.124 | 4.20M | 127 | 337 | 26.69 | 0.890 | 0.151 | 3.12M | 130 |
| FastGS | 150 | 24.56 | 0.848 | 0.210 | 0.73M | 609 | 133 | 24.71 | 0.818 | 0.252 | 0.61M | 578 |
| FastGS-Big | 227 | 25.41 | 0.877 | 0.166 | 1.52M | 511 | 256 | 25.81 | 0.861 | 0.190 | 1.41M | 433 |
| **TurboGS (Ours)** | 112 | 25.27 | 0.830 | 0.225 | 0.99M | 128 | 102 | 25.77 | 0.818 | 0.244 | 0.85M | 126 |
| **TurboGS-Big (Ours)** | 184 | 26.24 | 0.883 | 0.159 | 2.61M | 86 | 245 | 26.85 | 0.874 | 0.170 | 2.39M | 88 |

*Table 12.* **Scene-wise quantitative results on representative sub-regions of the Rubble (Turki et al., 2022) dataset at 4K resolution,** where COLMAP (Schönberger & Frahm, 2016) is independently performed for each sub-region to ensure fair evaluation.

| Method | Sub-Region #1 (view 172 to 228) | | | | | | Sub-Region #2 (view 58 to 114) | | | | | | Sub-Region #3 (view 690 to 746) | | | | | |
|---|---|---|---|---|---|---|---|---|---|---|---|---|---|---|---|---|---|---|
| | Time↓ | PSNR↑ | SSIM↑ | LPIPS↓ | $N_{GS}$ ↓ | FPS↑ | Time↓ | PSNR↑ | SSIM↑ | LPIPS↓ | $N_{GS}$ ↓ | FPS↑ | Time↓ | PSNR↑ | SSIM↑ | LPIPS↓ | $N_{GS}$ ↓ | FPS↑ |
| Taming-3DGS | 858 | 25.34 | 0.737 | 0.393 | 1.00M | 53 | 925 | 24.81 | 0.736 | 0.399 | 0.94M | 51 | 911 | 26.18 | 0.728 | 0.409 | 0.86M | 50 |
| FastGS | 639 | 26.17 | 0.791 | 0.322 | 1.90M | 87 | 630 | 25.79 | 0.799 | 0.290 | 3.73M | 141 | 574 | 26.51 | 0.775 | 0.315 | 3.05M | 149 |
| **TurboGS (Ours)** | 361 | 26.57 | 0.776 | 0.295 | 2.83M | 50 | 367 | 25.88 | 0.775 | 0.273 | 2.78M | 47 | 353 | 26.73 | 0.760 | 0.285 | 2.44M | 48 |

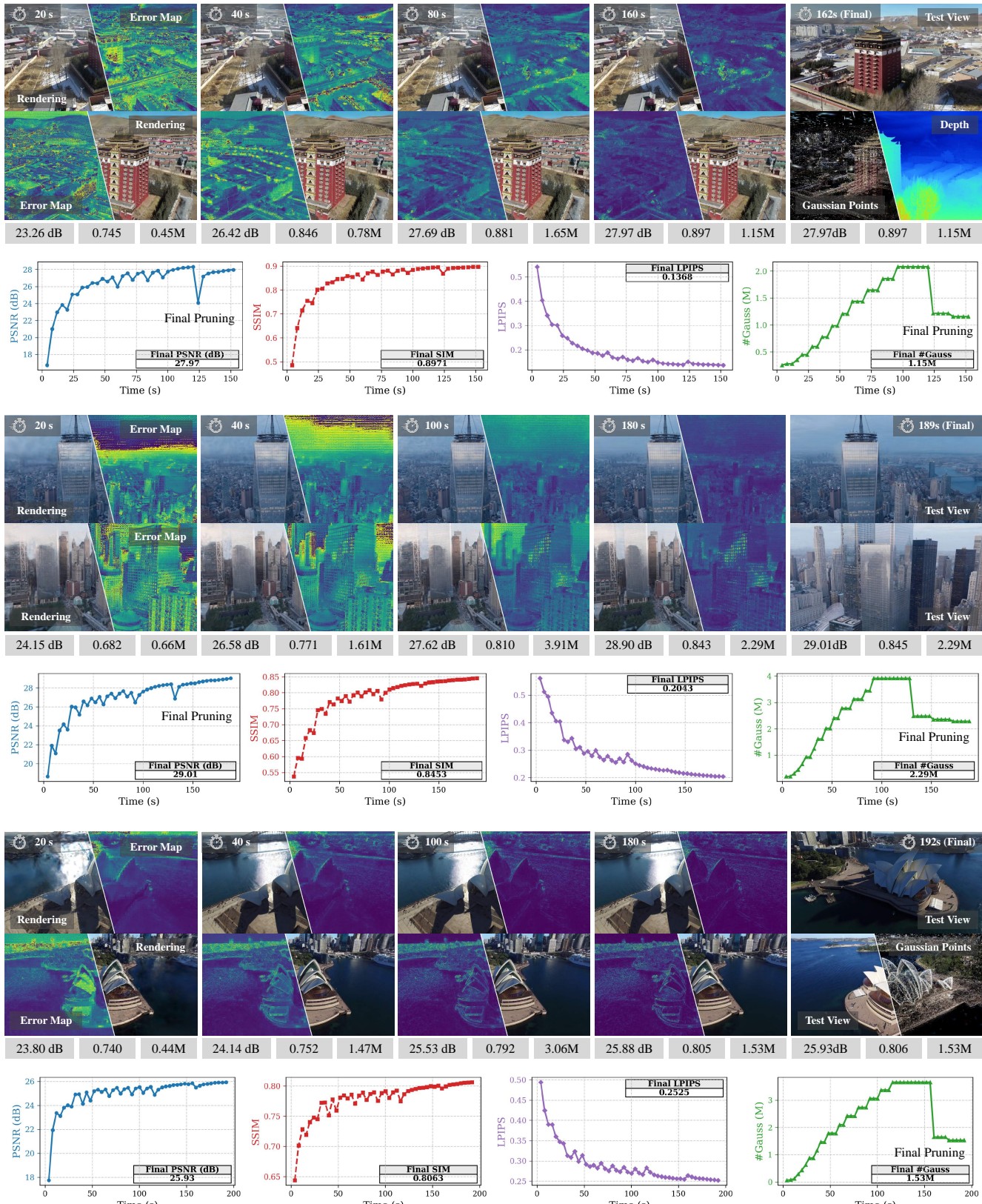

*Figure 14.* **Optimization progress on large-scale OMMO scenes.** We visualize rendering results, error maps, and training curves at representative iterations (*e.g.*, 20s, 40s, 80s, and final), showing progressive reconstruction improvement and convergence behavior.

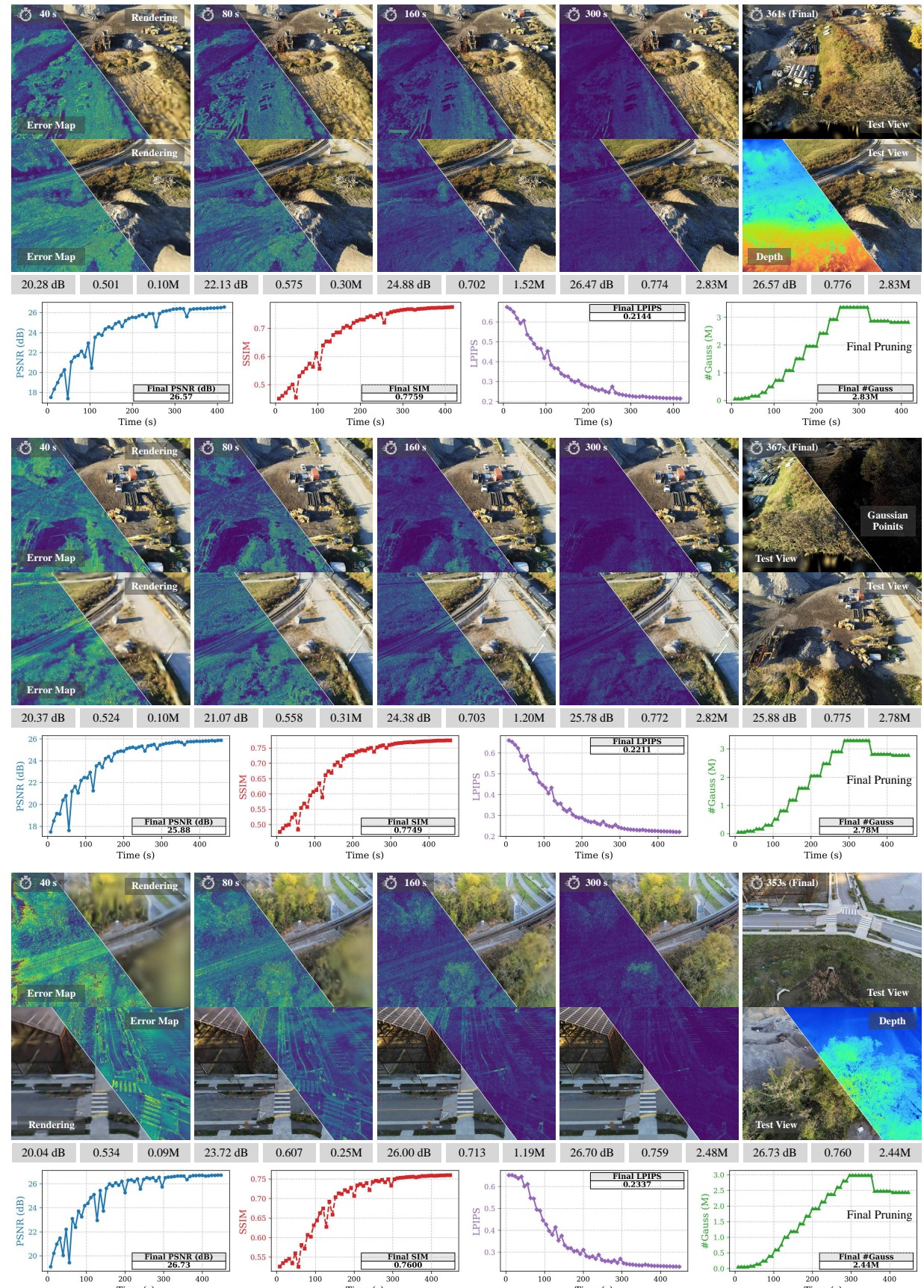

*Figure 15.* **Optimization progress on 4K Rubble sub-regions.** Representative rendering results and training curves during optimization.

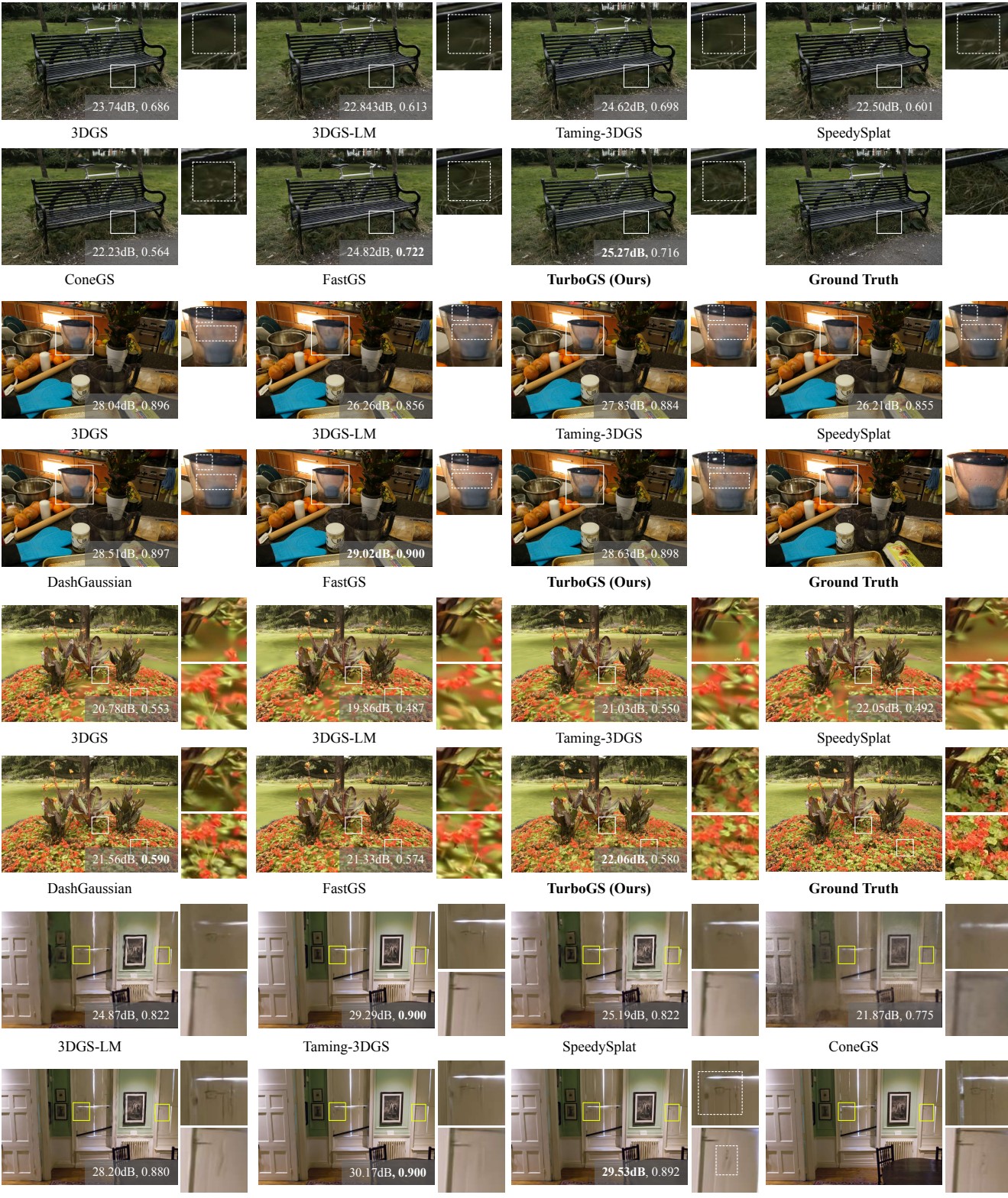

*Figure 16.* **Additional qualitative comparisons under the same training time budget.** We visualize representative results from existing fast 3DGS optimization methods at the time when TurboGS finishes optimization, on the Mip-NeRF 360 (Barron et al., 2022) dataset (*Bicycle*, *Counter*, *Flowers*) and the Deep Blending (Hedman et al., 2018) dataset (*Drjohnson*). Cropped patches highlight differences in detail reconstruction and visual fidelity.

