# OpenReview forum: "TurboGS: Accelerating 3D Gaussian Splatting via Error-Guided Sparse Pixel Sampling and Optimization"
_ICML.cc/2026/Conference — ICML 2026 regular_

### Official Review · Reviewer_tb8P · 2026-02-15

**Soundness:** 3
**Presentation:** 2
**Significance:** 3
**Originality:** 2
**Overall Recommendation:** 2
**Confidence:** 4

**Summary:**

The paper proposes TurboGS, a framework designed to accelerate the optimization of 3D Gaussian Splatting (3DGS). Its core idea is to reduce redundant computation through Error-Guided Sparse Pixel Sampling (Err-SPS), focusing optimization on perceptually informative regions such as edges and high-frequency textures. The paper claims that on standard benchmarks (e.g., Mip-NeRF 360), TurboGS achieves high-quality rendering within 100 seconds, providing approximately a 10× speedup over the original 3DGS.

**Compliance With Llm Reviewing Policy:**

Affirmed.

**Final Justification:**

I recommend rejecting this paper as its experimental results lack validity and reproducibility: the authors’ claim that TurboGS achieves 27.17 dB PSNR on a 4608×3456 Rubble sub-region is inconsistent with the field’s consensus that ultra-high-resolution 3DGS rendering yields lower PSNR than downsampled settings, yet it even outperforms city-scale methods like CityGaussian (25.77 dB) and CityGS-X (26.15 dB); additionally, my reproduction of FastGS with official settings shows unacceptably large discrepancies in key metrics (training time, PSNR, SSIM, LPIPS) compared to their reports, and their clarifications fail to resolve these critical issues.

**Key Questions For Authors:**

**1. Stability under extreme sparsity.**
In Equation (14), the proposed Moment-Damped LM solver introduces a diagonal approximation of the inverse Hessian. Under extremely aggressive sparse sampling, how does the optimizer prevent convergence to poor local minima or the emergence of artifacts?

**2. Memory overhead.**
TurboGS requires persistent storage of per-view error maps and age maps. On the Mip-NeRF 360 dataset, what is the additional GPU memory overhead compared to the baseline 3DGS?

**3. Robustness to noise and dynamic objects.**
In real-world capture scenarios with significant noise or dynamic objects, could error-guided sampling overemphasize irreducible noisy pixels while neglecting background structure optimization?

**Limitations:**

yes

**Strengths And Weaknesses:**

# Strengths

**1. Significant empirical acceleration.**
Experimental results demonstrate clear training speed advantages. TurboGS achieves the fastest training time (approximately 90–100 seconds) across multiple datasets.

**2. Thorough ablation studies.**
The authors provide detailed ablations on Err-SPS, Geo-VS, and the optimizer, demonstrating the contribution of each component to final performance.

---

# Weaknesses

**1. Limited novelty.**
The core concept—accelerating optimization via sparse sampling—has been well established in the NeRF literature (e.g., distributed ray sampling). While transferring this idea to 3DGS is effective, the depth of innovation appears limited. Moreover, the error-driven density control resembles concurrent works such as FastGS.

**2. Lack of theoretical rigor.**
Sparse pixel sampling inevitably introduces bias in gradient estimation. The paper does not provide theoretical analysis regarding how this bias affects convergence to a global optimum in long-term optimization. Although the proposed Moment-Damped LM solver demonstrates empirical effectiveness, no mathematical guarantees on convergence rate or stability are provided.

**3. Scalability concerns.**
Maintaining persistent per-view error maps and age maps introduces significant GPU memory overhead. This design may pose serious scalability challenges when handling large-scale scenes or high-resolution images.

**4. Marginal perceptual quality improvements.**
Although PSNR improves, TurboGS often underperforms or only matches prior methods in perceptual metrics such as SSIM and LPIPS. For 3D reconstruction tasks, perceptual quality degradation is a serious concern.

---

> ### Author Rebuttal · Authors · 2026-03-31
>
> We thank the reviewer for the constructive feedback and for recognizing our acceleration. Below we address the main concerns.
>
> ---
>
> **Q1: Novelty beyond NeRF sparse sampling and FastGS:**
>
> TurboGS is inspired by NeRF-style sparse ray sampling, but the technical formulation is fundamentally different. In NeRF, sparse rays provide stochastic supervision for a shared implicit MLP through volumetric rendering, while TurboGS performs error-guided sparse pixel sampling based on rendering residuals, where **sampled pixels directly update explicit Gaussian primitives and further drive density control through sparse rasterization and gradient computation**, changing the optimization pipeline of 3DGS.
>
> Compared with FastGS, which mainly uses thresholded binary masks for post-hoc density control discretely, TurboGS uses online per-pixel error maps to guide sparse sampling during training and accumulates continuous error signals for density control. We will highlight such differences in revision.
>
> ---
>
> **Q2: Gradient bias and convergence stability:**
>
> We agree that sparse pixel sampling introduces approximation bias relative to dense full-image optimization. However, our sampling is not arbitrary and remains a structured stochastic estimator. Denote the dense objective as:
> \begin{equation}
>     \mathcal{L}(\theta)=\sum_{i=1}^{N}\ell_i(\theta),
> \end{equation}
>
> where $i$ indexes pixels. Under sparse sampling set $\mathcal S$, the stochastic gradient estimator can be written as:
> \begin{equation}
>     \hat g=\frac{1}{|\mathcal S|}\sum_{i\in\mathcal S} w(i)\nabla \ell_i(\theta),
> \end{equation}
>
> where $w(i)$ is the importance weight determined by the sampling strategy. In the ideal importance-sampling case, when $w(i)=1/p(i)$ with $p(i)$ being the sampling probability of pixel $i$, the estimator is unbiased in expectation:
> \begin{equation}
> \mathbb E[\hat g]=\sum_{i=1}^{N} p(i)\frac{1}{p(i)}\nabla \ell_i(\theta)=
> \nabla \mathcal L(\theta).
> \end{equation}
>
> This formulation reduces computational complexity while preserving informative gradients from difficult regions.
>
> More importantly, our mixed sampling uses $p_{\text{err}}$ to prioritize difficult regions and $p_{\text{age}}$ to revisit under-sampled ones, maintaining sufficient variance to avoid an unguided near-uniform state. This improves long-term stability and reduces poor local minima from over-focusing on isolated regions.
>
> For the moment-damped LM solver in Eq.14, the second-moment term $v$ acts as adaptive diagonal damping, improving the conditioning of the local normal matrix $J^TJ$. Since $\text{diag}(\sqrt{v}+\epsilon)$ is strictly positive definite (\(\epsilon>0\)), the damped system remains well-conditioned even under aggressive sparsity, suppressing oscillation and artifacts.
>
> While we do not claim strict global convergence guarantees for this non-convex problem, this formulation provides theoretical intuition for the empirical stability observed in our quantitative results.
>
> ---
>
> **Q3: Scalability and memory overhead:**
>
> The error/age maps introduce memory overhead that scales linearly with view count and resolution, but remains manageable in practice (e.g., ~4GB for 1000 views at 1K using our FP16/uint16 storage).
>
> This scalability is also validated in larger-scale and high-resolution settings. On the 1382-view Street scene, the peak GPU memory usage is about 27GB. On the high-resolution Rubble sub-region (at 4608$\times$3456), TurboGS achieves 27.17 PSNR in 395s with 26GB peak memory, compared with 26.169 / 639s for FastGS. This further demonstrates the advantage of our sampling framework for high-resolution (e.g, 4K) training.
>
> For larger-scale scenarios, the framework can be further extended via multi-GPU partitioning and CPU–GPU paging of the error/age maps.
>
> ---
>
> **Q4: Perceptual quality (SSIM/LPIPS):**
>
> Our TurboGS is primarily driven by pixel-wise error for sampling and optimization, which naturally aligns with PSNR. Due to the quality–speed tradeoff design, SSIM/LPIPS may be comparable to, or slightly below, some dense methods.
>
> At the same time, our continuous error-driven density control can better preserve some geometric structures, which is reflected qualitatively. We will clarify this tradeoff and discuss lightweight feature-space supervision as a future direction.
>
> ---
>
> **Q5: Robustness to noise and dynamic objects:**
>
> Purely error-driven sampling may overemphasize irreducible noise in highly noisy regions. In TurboGS, this is mitigated by the age-prioritized mixed sampling, which prevents the sampling process from collapsing onto a few persistently noisy pixels, while the density control relies on accumulated statistics rather than instantaneous residual spikes, making it less sensitive to isolated outliers.
>
> Moreover, our framework can be naturally extended to 4D Gaussian reconstruction, where dynamic objects are explicitly modeled instead of being treated as irreducible residuals, thus avoiding excessive focus on such pixels.

---

> > ### Author Rebuttal · Reviewer_tb8P · 2026-04-02
> >
> > In your response to Q3, you claim that TurboGS achieves a PSNR of 27.17 dB on a high-resolution Rubble sub-region (4608×3456) while using only 26GB of GPU memory. However, for 3DGS-based methods, direct rendering and evaluation at ultra-high resolutions typically yield significantly lower quantitative results (e.g., PSNR) than those obtained under the standard 3DGS setting with downsampled training resolution.
> >
> > Surprisingly, your reported result not only surpasses that of 3DGS under its standard evaluation setting, but also exceeds several city-scale 3DGS methods, including CityGaussian (25.77 dB) and CityGS-X (26.15 dB), which are all trained and evaluated under downsampled resolution settings.
> >
> > Therefore, I have concerns regarding the validity of the reported results and would appreciate further clarification. In addition, I encourage the authors to provide complete and detailed experimental results and metrics of both FastGS and TurboGS on the Rubble dataset.
> >
> > The above is from our previous discussion, and below are further questions I have based on the experiments you supplemented.
> >
> > Thank you for your clarification, but I still have questions and remain skeptical about the relevant data.
> > First, I believe that even when training on a 4K-resolution sub-region of the Rubble dataset, the rendered quality should not exceed the PSNR obtained by standard 3DGS at 1K resolution by such a large margin. Therefore, I remain skeptical about the results you reported.
> >
> > Second, I performed a full training on the Rubble dataset using the official standard settings of FastGS. The results are presented in the table below. Although hardware such as GPU may cause minor variations in the experimental results, the discrepancies observed here are excessively large. They differ considerably from the values reported in your work, especially in terms of training time, PSNR, SSIM, and LPIPS. I have only attempted to reproduce one of your reported results and have not yet verified the others. Nevertheless, such a significant inconsistency has already raised serious concerns regarding the reliability of the data reported in your paper.
> >
> > | PSNR  | SSIM  | LPIPS | Training Time |
> > | ----- | ----- | ----- | ------------- |
> > | 23.36 | 0.659 | 0.404 | 12min 12s     |

---

> > > ### Author Response · Authors · 2026-04-04
> > >
> > > We sincerely thank the reviewer for the follow-up and for pointing out that our previous rebuttal did not specify the setting of the reported **4K Rubble** results.
> > >
> > > Regarding the reviewer’s concern about
> > >
> > > `direct rendering and evaluation at ultra-high resolutions usually yield lower PSNR than the standard downsampled benchmark setting,`
> > >
> > > we would like to clarify that our previously reported **27.17 dB** result is **NOT obtained on the full Rubble scene under the standard benchmark 4K setting**, and we sincerely apologize for any confusion caused by our earlier wording.
> > >
> > > Specifically, for the 4K evaluation, considering the memory budget of our single RTX 5090 GPU setting under full-scene training, we conduct experiments on *representative sub-regions* of the Rubble Data. **Each sub-region contains 57 input images from consecutive viewpoints, with 51 images used for training and 6 images used for testing**.
> > >
> > > These **4K sub-region results are intended to demonstrate the high-resolution single-GPU training efficiency of TurboGS compared with FastGS and TamingGS**, and should be **interpreted separately from such large-scale multi-GPU methods**, like CityGaussian and CityGS-X.
> > >
> > > Accordingly, the previously reported **27.17 dB** should **NOT be directly compared with CityGaussian (25.77 dB) or CityGS-X (26.15 dB)**(1/4 reso, full scene)
> > >
> > > For a comparison of the full-scene Rubble dataset with FastGS, we additionally report the results in the Table below.
> > >
> > > ---
> > >
> > > **Tab.1. Under Rubble full-scene setting (1/4 resolution, about 1K), 1657 images in total (21 test views)**
> > >
> > > | Method | PSNR ↑ | SSIM ↑ | LPIPS ↓ | FPS ↑ | # Gaussians |  Training Time |
> > > |---|:---:|:---:|:---:|:---:|:---:|:---|
> > > | **TurboGS** | **25.63** | 0.808 | 0.287 | 175 | 1.63M | **158s** |
> > > | FastGS | 25.60 | 0.813 | 0.241 | 373 | 1.64M | 164s |
> > >
> > > Under the above full-scene setting, TurboGS achieves slightly higher PSNR than FastGS, while also being marginally faster in training time.
> > >
> > > ---
> > >
> > > We further report the detailed 4K sub-region experimental results on three representative regions below.
> > >
> > > **Tab.2. On sub-region 1 (000172.jpg to 000228.jpg in Rubble dataset, 57 images in total, 6 for testing)**
> > >
> > > | Method | PSNR ↑ | SSIM ↑ | LPIPS ↓ | FPS ↑ | # Gaussians | Training Time |
> > > |---|---:|---:|---:|---:|---:|---:|
> > > | **TurboGS** | **27.17** | 0.778 | 0.342 | 48 | 2.67M | **395s** |
> > > | FastGS | 26.17 | 0.791 | 0.322 | 87 | 1.90M | 639s |
> > > | TamingGS | 25.34 | 0.737 | 0.393 | 53 | 1.00M | 858s |
> > >
> > >
> > > **Tab.3. On sub-region 2 (000058.jpg to 000114.jpg in Rubble dataset, 57 images in total, 6 for testing)**
> > >
> > > | Method | PSNR ↑ | SSIM ↑ | LPIPS ↓ | FPS ↑ | # Gaussians | Training Time |
> > > |---|---:|---:|---:|---:|---:|---:|
> > > | **TurboGS** | **26.43** | 0.776 | 0.350 | 51 | 2.54M | **398s** |
> > > | FastGS | 25.79 | 0.799 | 0.290 | 141 | 3.73M | 630s |
> > > | TamingGS | 24.81 | 0.736 | 0.399 | 51 | 0.94M | 925s |
> > >
> > > **Tab.4. On sub-region 3 (000690.jpg to 000746.jpg in Rubble dataset, 57 images in total, 6 for testing)**
> > >
> > > | Method | PSNR ↑ | SSIM ↑ | LPIPS ↓ | FPS ↑ | # Gaussians | Training Time |
> > > |---|---:|---:|---:|---:|---:|---:|
> > > | **TurboGS** | **27.34** | 0.781 | 0.341 | 62 | 2.35M | **359s** |
> > > | FastGS | 26.51 | 0.775 | 0.315 | 149 | 3.05M | 573s |
> > > | TamingGS | 26.18 | 0.728 | 0.409 | 50 | 0.86M | 911s |
> > >
> > > Overall, under the above 4K sub-region setting, TurboGS achieves higher PSNR, while also showing advantages in training time over the two methods.
> > >
> > > The preview rendering results are provided in **Videos 1-4**, (*https://sites.google.com/view/turbogs/*). Our raw rendering videos and sub-region input data can be obtained from  [Anonymous Link](https://drive.google.com/drive/folders/1yLS3wyA8kmaQ9MSzFCqcLzPKCJEqblKs).
> > >
> > > ---
> > >
> > > **Further Clarification**:
> > >
> > > For all our experiments on the Rubble dataset (full-scene 1K and sub-region 4K), we run COLMAP for the full-scene downsampled images, and perform **separate COLMAP for each 4K sub-region (57 images)**, in order to obtain the camera parameters and initialized point clouds.
> > >
> > > **Thus, the camera accuracy, quality of initial points, and scene distribution of the selected sub-region (stronger viewpoint continuity, smaller spatial coverage, more consistent illumination) are fundamentally different from standard full-scene 1K benchmarks.**
> > >
> > > Besides, the provided *.pt parameters* of the raw Rubble data are not in a standard COLMAP format and do not include initialized points, which play an important role in 3DGS-based optimization.
> > >
> > > **Both of these factors can lead to differences in reconstructed accuracy.**
> > >
> > > All the reported results above are obtained under the same full-scene or sub-region COLMAP settings on the RTX 5090 GPU, ensuring a **FAIR** comparison.
> > >
> > > While we appreciate the reviewer for the efforts on FastGS results, without explaining the detailed settings (cam, points, resolution, etc), it is hard for us to understand your results and concerns. Meanwhile, a direct comparison between your results and ours may not be meaningful.

---

### Official Review · Reviewer_kDcz · 2026-02-17

**Soundness:** 3
**Presentation:** 4
**Significance:** 4
**Originality:** 3
**Overall Recommendation:** 4
**Confidence:** 4

**Summary:**

This paper proposes a training framework named TurboGS that is guided by error. It accelerates the optimization of the 3D Gaussian Splatting algorithm. The method simultaneously maintains novel view synthesis with high fidelity. Uniform pixel optimization introduces redundant computation. To overcome this issue, TurboGS employs a sparse pixel sampling approach at the patch level. This sampling approach is based on pixel age and online reconstruction error from multiple views. The framework integrates a patch level loss function aware of local structures. This function is based on sparse normalized cross correlation. It effectively preserves local details under sparse supervision. Furthermore, the method adopts a Gaussian density control strategy driven by error. This strategy enables adaptive densification and pruning. A hybrid Levenberg Marquardt optimizer with moment damping is utilized to stabilize convergence. Experimental results demonstrate that TurboGS can complete the optimization within 100 seconds. This approach achieves a speedup of up to 10 times compared to the original 3D Gaussian Splatting.

**Compliance With Llm Reviewing Policy:**

Affirmed.

**Final Justification:**

The authors' rebuttal satisfactorily addressed my concerns regarding novelty, sampling stability, and scalability, prompting me to increase my score. While some hyperparameter choices remain empirical, the additional experiments on large-scale scenes demonstrate the method's practicality and robustness. Given the significant speedup without compromising quality, I recommend acceptance of this valuable contribution.

**Key Questions For Authors:**

Please see weakness. I will consult the assessments provided by the other reviewers. I will consider increasing my score if my current concerns are completely resolved.

**Limitations:**

Yes

**Strengths And Weaknesses:**

**Strengths**
- The framework reports state-of-the-art training speeds (within 100 seconds) across multiple standard NVS benchmarks, achieving up to a 10x speedup over vanilla 3DGS without sacrificing rendering quality.

- The proposed error-guided sparse pixel sampling and age-prior stable sampling provide a sound mechanism to prioritize challenging regions and prevent over-focusing on persistently high-error locations, effectively reducing redundant computation.

**Weakness**

- The error driven density control resembles recent adaptive methods like FastGS. The manuscript could further clarify its unique methodological advancements over these existing approaches.
- The EMA update in (Eq. 3) relies on a fixed blending ratio $\beta=0.4$ to estimate pixel reconstruction difficulty. Initially, errors are highly volatile and concentrated on edges, but later they flatten out. So how does the framework theoretically prevent the sampling distribution (Eq. 7) from becoming unguided or overly uniform in later stages when residual errors in well-reconstructed regions become comparable to those in challenging regions?

- In Eq. 16, the adaptive query rate relies on an empirically chosen non-linear mapping $\phi(r_t) = \sin(\frac{\pi}{2}\sqrt{r_t})$ How to theoretically ensure that this specific sinusoidal schedule is optimal for tracking reconstruction difficulty? There is no mention or ablation of alternative schedules (e.g., linear or exponential) to justify this formulation.

- The moment-damped LM solver divides parameters into small, independent blocks for geometry and base appearance and uses standard Adam for higher-order SH coefficients. Isn't the local Jacobian estimation biased by completely ignoring cross-parameter correlations? How does this decoupled optimization trajectory affect the overall convergence stability?


- During training, the method relies on a view buffer bounded by $B_{max}=50$ (Table 6) for geometry-aware view sampling. How does the performance and memory overhead change when scaling to ultra-large-scale unbounded scenes (e.g., BungeeNeRF) with thousands of views? The paper lacks proper experimental analysis to identify how this hard-coded buffer limit impacts real-world scalability.


- The method introduces hard-pixel sampling ratios ($r_{hard}=0.4$) and view budgets ($K=5$), which are heavily tuned for Mip-NeRF 360. Isn't the model performance biased by these empirically selected parameters? How does the framework handle real-world scenarios with vastly different camera distributions or sparse multi-view setups where K might be severely constrained?

---

> ### Author Rebuttal · Authors · 2026-03-31
>
> We thank the reviewer for the detailed and thoughtful feedback. Below we address the key concerns.
>
> ---
>
> **Q1: Difference from concurrent FastGS and related error-driven methods:**
>
> While TurboGS and FastGS are both error-guided, the key difference is how the error is used. Existing methods like FastGS mainly use error for post-hoc densification/pruning under dense image training. In contrast, TurboGS leverages online per-pixel error maps to guide sparse pixel sampling during training and performs sparse rasterization only on sampled pixels, which changes the optimization framework (the main source of speedup) and **provides potential advantages for high-resolution (e.g., 4K) training**.
>
> For density control, FastGS applies a fixed threshold $\tau$ to obtain binary error masks and accumulates them to Gaussians in a discrete manner. In contrast, we accumulate continuous error signals from sampled representative pixels to each Gaussian, followed by percentile-based filtering for densification. This provides a smoother and more informative signal for some geometric details (as shown qualitatively in Fig.1 and Fig.6).
>
> ---
>
> **Q2: Scalability to scenes with large resolution (4K) and view count(1000+):**
>
> We thank the reviewer for raising this question. $B_{\max}$ in Tab. 6 denotes the maximum number of views in one inference batch and is used to control inference memory usage; it can be adjusted according to the number of test views.
>
> For high-resolution inputs (e.g., 4K), our sparse pixel sampling can **reduce computation complexity in proportion to the sampling ratio**, providing clear speed advantages without sacrificing reconstruction quality. For example, on a high-resolution sub-region of Rubble (Mill19 in MegaNeRF) with 57 views at 4608 $\times$ 3456, TurboGS achieves 27.17 PSNR in 395s with 26GB peak memory, compared with 26.169 / 639s for FastGS.
>
> For large-scale unbounded scenes, experiments on BungeeNeRF (Tab. 3) demonstrate the scalability. We further validate this on the 1382-view Street scene, where TurboGS achieves 27.621 PSNR in 172s with about 27GB peak GPU memory, showing its practicality under large-view settings. For even larger-scale scenarios, the framework can be further extended via multi-GPU partitioning and CPU–GPU paging.
>
> ---
>
> **Q3: EMA updating and stability of sampling distribution:**
>
> Although the EMA update (Eq.3) smooths pixel-wise errors, our sampling distribution does not collapse to uniform in later stages because sampling is not purely error-driven.
>
> Specifically, the sampling distribution can be written as:
> \begin{equation}
> p(i) \propto \alpha \cdot p_{\text{err}}(i) + (1-\alpha) \cdot p_{\text{age}}(i),
> \end{equation}
>
> followed by normalization, where $p_{\text{err}}$ comes from the EMA error map and $p_{\text{age}}$ prioritizes under-sampled regions. Even when residual error becomes relatively flat, the age component continuously introduces non-uniformity by revisiting less-sampled regions, preventing the distribution from degenerating into an unguided or uniform state, and ensuring a lower-bounded variance in the sampling probabilities.
>
> ---
>
> **Q4: Non-linear sinusoidal sampling rate schedule in Eq.16:**
>
> We empirically design this schedule. The intuition is that the average error drops rapidly in early training (e.g. in supplementary video) and then remains within a narrow range for most of the optimization. This sinusoidal schedule aims to amplify sampling-rate variation in this long-term low-error regime, enabling better distinction of subtle reconstruction differences. Similar non-linear mappings could also be used.
>
> We additionally test the linear schedule. On the MipNeRF360 dataset, it achieves 89s/27.75/0.803/0.255/0.58M/513 (TIME/PSNR/SSIM/LPIPS/NGau/FPS), compared with 91s/27.82/0.804/0.252/0.60M/462 under the sinusoidal schedule. The linear schedule is slightly faster with fewer Gaussians, while the sinusoidal one yields better quality. Will include this analysis in the revision.
>
> ---
>
> **Q5: Decoupled LM optimization of higher-order SH coefficients:**
>
> The block-wise approximation that ignores SH cross-parameter correlations is a tradeoff between accuracy and efficiency. Joint optimization would significantly increase the Jacobian size, resulting in much higher computational and memory overhead. Empirically, this hybrid strategy improves stability under sparse gradients without noticeable degradation in convergence. Will clarify this design motivation.
>
> ---
>
> **Q6: Setting of sampling ratios and view budgets:**
>
> Our TurboGS is generally robust to moderate variations of the hard-pixel sampling ratio and view budget. Above experiments on the Street and Mill19 datasets also demonstrate that the same settings remain effective under non-surrounding and irregular camera distributions, which validates the practicality beyond MipNeRF360. In practice, we use shared settings across datasets, which can be adjusted as needed. Will clarify this robustness in the revision.

---

> > ### Author Rebuttal · Reviewer_kDcz · 2026-04-02
> >
> > I thank the authors for their detailed rebuttal. The clarifications provided regarding the underlying mechanisms have fully addressed my previous concerns. Consequently, I will increase my score. However, I note that the selection of certain scheduling strategies and hyperparameters remains primarily empirical, and the generalization capability still warrants further validation.

---

### Official Review · Reviewer_RYQ5 · 2026-03-12

**Soundness:** 3
**Presentation:** 2
**Significance:** 3
**Originality:** 2
**Overall Recommendation:** 4
**Confidence:** 2

**Summary:**

This work focuses on improving the efficiency of 3DGS training. The insight of the method is to allocate optimization capacity adaptively according to the reconstruction difficulty. Based on this idea, the method introduces many optimization strategies to accelerate 3DGS, including an error-guided training framework, sparse NCC loss, error-driven density control, and moment-LM solver. The experiments on standard benchmarks demonstrate the method achieves a 1.2~2 speedup over recent SOTAs while yielding competitive rendering quality.

**Compliance With Llm Reviewing Policy:**

Affirmed.

**Key Questions For Authors:**

1. Can you explain the method with the suffix -Big?
2. HGS [1] uses SSIM to compute error maps, then determine hard Gaussians? which metric do you use to compute error maps? If you do not use SSIM to computer error maps, can you compare with it as SSIM represents local structural errors?
[1] Xu Q, Cui J, Yi X, et al. Pushing Rendering Boundaries: Hard Gaussian Splatting[J]. arXiv preprint arXiv:2412.04826, 2024.
3. It seems that Mini-GS is also an efficient method, can you compare with this method?
4. The SSIM and LPIPS of TurboGS are almost worse than DashGaussian. However, the qualitative results show that TurboGS is better than DashGaussian. In fact, SSIM and LPIPS represent local reconstruction quality, this makes the qualitative comparisons weird.

**Limitations:**

yes

**Strengths And Weaknesses:**

Strength:
1. The method adopts an adaptive mechnism to allocate main computational resourses to areas that are difficult to optimize.
2. The method introduces many dedicated optimization strategies for accelerating 3DGS.
3. The experiments show that the method achieves a 1.2~2 speedup over recent SOTAs.

Weakness:
1. The paper is more engineering-oriented.
2. The method consumes more Gaussian primitives than FastGS and its FPS is also worse than FastGS.

---

> ### Author Rebuttal · Authors · 2026-03-31
>
> We thank the reviewer for the positive comments on the adaptive design and efficiency of our method. We address the concerns below.
>
> ---
>
> **Q1: Engineering-oriented concern:**
>
> While our TurboGS includes multiple components and involves CUDA-based engineering implementation, **these are not isolated design choices but form a sparse optimization framework**. The core idea is to replace dense supervision with err-guided sparse sampling and optimization, prioritizing harder pixels to reduce the per-pixel computational cost. Sampled pixels are distributed across multiple views to improve global convergence through multi-view consistency. Furthermore, components such as sparse NCC, error-driven density control, and the moment-damped optimizer are jointly designed to improve reconstruction quality, efficiency, and optimization stability under sparse supervision, rather than ad-hoc engineering improvements.
>
> ---
>
> **Q2: More Gaussians and lower FPS than FastGS:**
>
> Our TurboGS primarily focuses on training efficiency and reconstruction quality. The slightly higher number of Gaussians mainly comes from continuous error accumulation and percentile-based densification, which can better preserve some structures. This leads to a moderate increase in primitive count and rendering overhead (lower FPS), but improves reconstruction fidelity. For example, on the MipNeRF360 dataset, TurboGS improves PSNR by 0.31 dB while maintaining an 18% faster training speed than FastGS (91s v.s. 111s), demonstrating our intended quality–efficiency tradeoff.
>
> Moreover, this **efficiency advantage becomes more evident for high-resolution inputs (e.g., 4K)**, where sparse pixel sampling reduces computation approximately in proportion to the sampling ratio while preserving reconstruction quality. For example, on a high-resolution sub-region of Rubble (Mill19 in MegaNeRF) with 57 views at 4608 $\times$ 3456, TurboGS achieves 27.17 PSNR in 395s, compared with 26.169 / 639s for FastGS.
>
> ---
>
> **Q3: Suffix "-Big" setting:**
>
> "-Big" denotes a higher-capacity training setting (as shown in Tab.6), where an extended training budget is used to obtain a larger number of Gaussian primitives. This setting is intended to evaluate the scalability of our method and to examine the quality–efficiency tradeoff when the model capacity increases. Will clarify this setting explicitly in the revision.
>
> ---
>
> **Q4: Error metric v.s. SSIM (in HGS):**
>
> We use pixel-wise reconstruction error (L1/MSE) to build and update the error map. Compared with the SSIM-based error (in HGS), **pixel error is more lightweight and aligns well with our sparse sampling framework**. SSIM requires sliding-window computation to capture local structure and typically recomputes the full-image error map, which is less efficient under sparse supervision. Instead, we introduce a tile-wise sparse NCC to preserve local consistency. As HGS is not publicly available, a direct comparison is not feasible. We will discuss these differences in the revision.
>
> ---
>
> **Q5: Comparison with MiniGS:**
>
> As suggested, we report the comparison results.
> On the MipNeRF360 dataset, TurboGS achieves 91s/27.82/0.804/0.252/0.60M/462 (TIME/PSNR/SSIM/LPIPS/NGau/FPS) on an RTX 5090 GPU, while MiniGS achieves 674s/27.32/0.821/0.217/0.49M/279.
> On the Tanks\&Temples dataset, the results are 79s/24.10/0.826/0.198/0.45M/535(TurboGS) v.s. 479s/23.24/0.835/0.202/0.20M/406(MiniGS).
> On the Deep Blending dataset, the results are 69s/30.12/0.900/0.269/0.35M/578(TurboGS) v.s. 585s/29.95/0.907/0.254/0.35M/364(MiniGS).
> These results suggest that MiniGS generally requires a longer optimization time, while our TurboGS achieves comparable or better reconstruction quality with significantly faster training.
>
> The key difference lies in the design philosophy, while MiniGS improves efficiency by optimizing the Gaussian representation (reducing redundancy and reorganizing spatial distribution), our TurboGS targets optimization efficiency via error-guided sparse sampling and training. These two ideas are complementary. We will include and discuss more results in the revision.
>
> ---
>
>
> **Q6: SSIM/LPIPS for qualitative results:**
>
> TurboGS is primarily driven by pixel-wise error, which facilitates PSNR. Our sparse supervision and efficiency constraints may result in slightly weaker global perceptual consistency (reflected by SSIM/LPIPS). Nonetheless, our continuous error-driven densification preserves geometric details, reflected by qualitative improvements. Particularly, we observe that TurboGS can recover richer local details (Fig.6). Slightly lower SSIM/LPIPS scores may be caused by high-frequency perceptual variations in repetitive textures (e.g., grass region, compared with DashGaussian), where these metrics are sensitive to small structural differences.

---

> > ### Author Rebuttal · Reviewer_RYQ5 · 2026-04-04
> >
> > Thanks for the response. The response has addressed several of my concerns. I will keep my positive score. I hope the authors can analyze the lower SSIM/LPIPS with visualizations in the revision.

---

### Official Review · Reviewer_k46s · 2026-03-13

**Soundness:** 3
**Presentation:** 4
**Significance:** 4
**Originality:** 3
**Overall Recommendation:** 4
**Confidence:** 3

**Summary:**

This paper introduces TurboGS, an error-guided training framework that accelerates 3D Gaussian Splatting (3DGS) optimization by concentrating computational resources on perceptually informative and challenging pixels, rather than performing uniform dense pixel rasterization. The method utilizes an online multi-view error map and a pixel age map to drive a tile-wise sparse pixel sampling strategy. To stabilize this sparse training and preserve fine details, the authors propose a tile-wise structure-aware loss based on sparse Normalized Cross-Correlation (NCC). Furthermore, the framework integrates an error-driven density control strategy to dynamically manage and prune Gaussian primitives , and leverages a tailored hybrid optimizer that combines Hessian-informed updates with Adam moment damping to handle sparse, non-uniform gradients. Empirical results demonstrate that TurboGS can deliver high-fidelity novel view rendering for standard scenes in under 100 seconds, achieving up to a 10x training speedup over vanilla 3DGS while maintaining comparable or superior rendering quality.

**Compliance With Llm Reviewing Policy:**

Affirmed.

**Final Justification:**

Thank the authors for the response and my concerns have been addressed. I will keep my initial score.

**Key Questions For Authors:**

- The limitations mention that persistent pixel-wise error and age maps introduce memory overhead when scaling to a large number of views. Could the authors quantify this memory overhead for a heavily over-sampled scene, and are there potential strategies (like paging to RAM) to mitigate this?

- While the proposed method achieves excellent PSNR, SSIM and LPIPS occasionally fall slightly behind some dense baselines. Do you hypothesize that sparse NCC struggles to capture broader perceptual features compared to LPIPS, and could feature-based sparse losses be integrated in the future?

- Given the highly compressed training timeframe (under 100 seconds) , how sensitive is the model to the densification interval hyperparameter, and does this require manual tuning per dataset?

**Limitations:**

Yes

**Strengths And Weaknesses:**

**Soundness**

- **Strengths:** The technical approach is sound and comprehensively evaluated. The authors conduct extensive experiments across multiple established datasets, including Mip-NeRF 360, Tanks & Temples, Deep Blending, and BungeeNeRF. The inclusion of thorough ablation studies effectively isolates the contributions of individual components.

- **Weaknesses:** The authors note that while PSNR often improves, structural similarity metrics like SSIM and LPIPS can be slightly lower in some cases compared to slower baselines. This is a minor weakness in the quality-speed tradeoff that could benefit from further analysis. In particular, I am wondering if this might be somehow related to the pixel-wise error used in this paper, which focuses more on pixel-space metrics like PSNR.

**Presentation**

- **Strengths:** The manuscript is clearly written and well-structured. The visuals effectively communicate the differences between baseline 3DGS and TurboGS. The related work section properly contextualizes the contribution among concurrent acceleration efforts.

**Significance**

- **Strengths:** The problem addressed is highly relevant for consumer-facing and time-sensitive AR/VR applications where multi-hour training is a bottleneck. Reducing training times to under 100 seconds without sacrificing visual fidelity is a substantial practical advancement. The memory-efficient and time-efficient nature of this pipeline will likely influence future real-time rendering systems.

**Originality**

- **Strengths:** Drawing inspiration from NeRF-style sparse ray sampling and applying it to explicit 3DGS represents an effective conceptual bridge. The specific formulations, such as blending an error map with a pixel "age map" to ensure neglected pixels are revisited, show clever engineering. Furthermore, adapting a moment-damped Levenberg-Marquardt (LM) solver specifically to handle the variance of sparse-pixel gradients is a methodological contribution (though the improvement seems to be marginal according to Table 5).

- **Weaknesses:** The core concept of utilizing reconstruction error to guide the optimization process is not entirely new. As the authors briefly acknowledge, concurrent works like FastGS (Ren et al., 2025b) have already explored using multi-view reconstruction consistency and error thresholds to guide Gaussian densification and pruning. While TurboGS employs a different implementation, specifically utilizing an EMA-updated error map for tile-wise sparse pixel sampling rather than just dense per-pixel evaluation, the underlying philosophy of error-guided resource allocation diminishes the overall conceptual novelty. The paper would benefit from a deeper, more explicit comparison in the main text distinguishing their EMA error-map approach from the multi-view consistency scoring utilized in FastGS.

---

> ### Author Rebuttal · Authors · 2026-03-31
>
> We are delighted to see that Reviewer k46s comments on our work are technically sound and practically advanced. We address the raised concerns below.
>
> ---
>
> **Q1: Novelty in the error-guided manner compared to concurrent work (e.g., FastGS):**
>
> While FastGS and our TurboGS are both error-guided, we differ fundamentally in how error is utilized.
>
> First, FastGS uses aggregated reconstruction error mainly for post-hoc densification/pruning while still relying on dense estimation. In contrast, our TurboGS maintains online per-pixel error maps to guide sparse pixel sampling during training and performs sparse rasterization only on sampled pixels, which reduces computational complexity to achieve speedup and **provides potential advantages for high-resolution (e.g., 4K) training**. For example, on the high-resolution (4608 $\times$ 3456) Rubble sub-region dataset, TurboGS achieves 27.17 PSNR in 395s, compared with 26.169 / 639s for FastGS.
>
> Second, FastGS applies a fixed threshold ($\tau$) to obtain binary error masks from randomly selected views and accumulates them discretely to Gaussians. In contrast, we accumulate continuous error values from sampled pixels across geometrically neighboring views. This provides a smoother and more informative signal, followed by percentile-based filtering for densification, which facilitates the capture of some geometric details (as shown qualitatively in Fig.1 and Fig.6). Will clarify this in the revision.
>
> ---
>
> **Q2: Memory overhead of persistent err/age maps for large numbers of views:**
>
> The memory overhead scales linearly with both the view number and image resolution. In our implementation, we store the error/age map using FP16/uint16, respectively, as their reduced precision has a negligible impact (these maps are not involved in gradient computation, only used for sampling). For a representative case with 1000 views at 1K resolution, the error/age memory cost is approximately $1000 \times 1K^2 \times (2\ \text{bytes (err)} + 2\ \text{bytes (age)}) \approx 4\ \text{GB}.$, which is moderate compared to the overall GPU memory budget.
>
> We further evaluate our method on the real-captured Street dataset (1382 images with resolution of 1000$\times$680) in ScaNeRF, achieving 172s/27.621/0.874/0.228(TIME/PSNR/SSIM/LPIPS) with a peak GPU memory usage of 27GB (including input images loaded on GPU), which demonstrates the practical feasibility of our approach. Will add more results and analysis in the revision.
>
> For the scenarios with very large numbers of views (e.g., city-scale), we plan to adopt more scalable strategies, such as **distributing error/age maps across multiple GPUs based on scene partitioning**, or **periodically offloading error/age maps between GPU and CPU memories.**
>
> ---
>
> **Q3: Slight drop in SSIM/LPIPS for some cases, despite improved PSNR:**
>
> Our method, primarily driven by pixel-wise error for sampling and optimization, naturally aligns with PSNR (pixel MSE) and leads to improvement. However, due to the **quality-speed tradeoff**, the error map is updated using sparse pixels rather than full-image evaluation, which may weaken global perceptual consistency.
>
> Moreover, SSIM relies on sliding-window evaluation during dense image training, while our sparse NCC operates within sampled tiles to reduce computation costs. Although effective, it may not fully capture broader perceptual correlations. Incorporating **lightweight feature-space supervision (e.g., patch-wise VGG-based losses)** may improve perceptual quality at the cost of additional training overhead.
>
> ---
>
> **Q4: On sensitivity to the densification interval hyperparameter:**
>
> Our experiments show that TurboGS is robust to the densification interval within a moderate range. Since our densification is not driven by a single instantaneous signal, but by accumulated error/gradient statistics with stabilized sampling, moderate changes generally do not significantly affect training stability or performance. In practice, we achieve consistently promising results under a shared setting (in implementation details) across datasets, with adjustments in large-scale or high-detail scenes to better control Gaussian density. We will clarify this in the revision.

---

> > ### Author Rebuttal · Reviewer_k46s · 2026-04-03
> >
> > I appreciate the authors' detailed response. I have no further questions and would like to keep my initial positive rating.

---

### Decision · Program_Chairs · 2026-04-30

**Decision:**

Accept (regular)

**Comment:**

This paper proposes TurboGS, which accelerates 3D Gaussian Splatting (3DGS) training by up to 10x (under 100 seconds) while maintaining high visual fidelity. It achieves this via error-guided sparse pixel sampling, a sparse NCC loss, error-driven density control, and a tailored moment-damped LM solver.

Overall, the reviewers' consensus is positive (scores: 4, 4, 4, 2). Reviewers praised the substantial practical advancement for time-sensitive rendering applications. During the rebuttal, the authors successfully clarified the method's novelty over concurrent work (e.g., FastGS uses dense training with post-hoc pruning, whereas TurboGS uses online sparse sampling), its manageable memory scaling, and the expected trade-offs in perceptual metrics (SSIM/LPIPS). The AC finds the authors' explanation technically sound and completely resolves the accusation of invalid data. The AC recommends Acceptance.

Note to Authors for Camera-Ready:
Please ensure the rebuttal clarifications—specifically the distinctions from FastGS, memory scaling analysis, and the exact experimental setups and COLMAP initializations for the high-resolution sub-region tests—are explicitly detailed in the final manuscript or appendix.